# The Art of War with *Pseudomonas aeruginosa*: Targeting Mex Efflux Pumps Directly to Strategically Enhance Antipseudomonal Drug Efficacy

**DOI:** 10.3390/antibiotics12081304

**Published:** 2023-08-09

**Authors:** Asiyeh Avakh, Gary D. Grant, Matthew J. Cheesman, Tejaswini Kalkundri, Susan Hall

**Affiliations:** School of Pharmacy and Medical Sciences, Griffith University, Gold Coast, QLD 4222, Australia; ossie.avakhhajishirkiaei@griffithuni.edu.au (A.A.); g.grant@griffith.edu.au (G.D.G.); m.cheesman@griffith.edu.au (M.J.C.); tejaswini.kalkundri@griffithuni.edu.au (T.K.)

**Keywords:** efflux pump inhibitor, RND multidrug efflux pump, *Pseudomonas aeruginosa* antimicrobial resistance, MexAB-OprM, MexXY-OprM

## Abstract

*Pseudomonas aeruginosa* (*P. aeruginosa*) poses a grave clinical challenge due to its multidrug resistance (MDR) phenotype, leading to severe and life-threatening infections. This bacterium exhibits both intrinsic resistance to various antipseudomonal agents and acquired resistance against nearly all available antibiotics, contributing to its MDR phenotype. Multiple mechanisms, including enzyme production, loss of outer membrane proteins, target mutations, and multidrug efflux systems, contribute to its antimicrobial resistance. The clinical importance of addressing MDR in *P. aeruginosa* is paramount, and one pivotal determinant is the resistance-nodulation-division (RND) family of drug/proton antiporters, notably the Mex efflux pumps. These pumps function as crucial defenders, reinforcing the emergence of extensively drug-resistant (XDR) and pandrug-resistant (PDR) strains, which underscores the urgency of the situation. Overcoming this challenge necessitates the exploration and development of potent efflux pump inhibitors (EPIs) to restore the efficacy of existing antipseudomonal drugs. By effectively countering or bypassing efflux activities, EPIs hold tremendous potential for restoring the antibacterial activity against *P. aeruginosa* and other Gram-negative pathogens. This review focuses on concurrent MDR, highlighting the clinical significance of efflux pumps, particularly the Mex efflux pumps, in driving MDR. It explores promising EPIs and delves into the structural characteristics of the MexB subunit and its substrate binding sites.

## 1. *Pseudomonas aeruginosa:* The Battle-Worn Strategist and Its Intricately Orchestrated Defense Weaponry

*Pseudomonas aeruginosa*, a Gram-negative pathogen known for its opportunistic and nosocomial nature, stands as a formidable adversary with its multidrug resistance (MDR) capabilities. It prevails as a frequent culprit behind nosocomial infections (NIs), notably pneumonia, and poses significant threats to immunocompromised individuals and those afflicted with structural lung conditions such as cystic fibrosis (CF) [1]. The pathogen, responsible for 10–11% of all NIs, has observed an alarming rise in prevalence over the past decade [2,3]. Disturbingly, mortality rates have soared to as high as 40%, especially among immunocompromised patients [4,5]. Infection with *P. aeruginosa* can occur in an either acute or chronic state. Acute infections typically manifest in patients with compromised respiratory epithelium, underlying chronic lung disease, or as a consequence of acute illness leading to respiratory failure and necessitating mechanical ventilation [1,6]. Patients with underlying lung diseases, such as CF, may experience a more chronic form of *P. aeruginosa* infection, characterized by persistent and prolonged infection. CF patients are particularly susceptible to both acute and chronic *P. aeruginosa* infections. Additionally, those with burns, sepsis, cancer, or traumatic injuries and those with chronic obstructive lung disease and mechanical ventilation-associated pneumonia, encompassing cases related to COVID-19, are also at risk of contracting *P. aeruginosa* infections. These infections are of particular concern in immunocompromised patients [1,7,8,9]. 

*Pseudomonas aeruginosa* strains exhibiting MDR or extensive drug resistance (XDR) are persistent nosocomial pathogens, with occurrence rates ranging from 11.5% to 24.7% and 9.0% to 11.2%, respectively, posing significant clinical complexities [10,11,12]. In 2017, the World Health Organization (WHO) recognized the inclusion of MDR-*P. aeruginosa* as one of the pathogens in the ESKAPE group and emphasized the critical requirement for innovative approaches in addressing the severe infections associated with these particular microorganisms [13]. *Pseudomonas aeruginosa* infections are challenging to treat due to the bacterium’s remarkable genetic mutability and the acquisition of resistance to multiple classes of antibiotics [14]. 

*Pseudomonas aeruginosa* has a genome size of approximately 6.3 to 7.0 million base pairs (Mb), which is larger than that of many well-studied bacteria such as *Escherichia coli* (4.6 to 5.5 Mb) and *Staphylococcus aureus* (2.6 to 2.8 Mb), and many others. This larger genome size indicates its genetic complexity and potential for diverse biological features [15,16,17]. The pathogen, much like a chameleon adapting to its environment, exhibits a remarkable repertoire of survival strategies. Through the utilization of multiple chromosomal determinants and intricate regulatory pathways, this bacterium can develop heightened levels of antimicrobial resistance (AMR) and effectively withstand diverse external stimuli, mirroring the adaptive nature of its reptilian counterpart [18]. *Pseudomonas aeruginosa* displays an ever-growing MDR against a wide range of antibiotics that include fluoroquinolones (FQs), aminoglycosides, and third and fourth-generation cephalosporins and β-lactams [19]. *Pseudomonas aeruginosa* distinguishes itself from other Gram-negative bacteria through its exceptional AMR profile. It employs distinctive mechanisms, including constitutive and inducible expression of efflux pumps (MexAB-OprM and MexXY, respectively) and inducible AmpC cephalosporinase expression, and exhibits low outer membrane (OM) permeability [20,21]. 

The MexAB-OprM efflux pump, produced constitutively, provides inherent resistance to a wide range of β-lactams and FQs, acting as a basal defense mechanism. On the other hand, the inducible MexXY expression confers intrinsic resistance specifically against aminoglycosides, further enhancing the bacterium’s adaptive response to antibiotic exposure [22]. Through whole-genome screening of mutant libraries, the “intrinsic resistome” in *P. aeruginosa* has emerged, comprising a diverse range of genes. This resistome exerts a significant influence on the antibiotic susceptibilities of the bacteria, highlighting its pivotal role in AMR [18,23,24,25,26]. These genes can include those involved in efflux pumps, which help to expel antibiotics from the cell, as well as other mechanisms that enhance the bacterium’s ability to survive in the presence of antimicrobial drugs [27]. 

In addition to the intrinsic resistome, *P. aeruginosa* develops AMR through chromosomal mutations [18]. Overexpression of chromosomal AmpC cephalosporinase, regulated by genes involved in cell wall recycling pathways, is a prevalent mutation-driven β-lactam resistance mechanism, observed in >20% of clinical *P. aeruginosa* strains [28,29,30]. In the context of various resistance mechanisms working in concert to develop antibiotic resistance, it is important to highlight the role of mutational inactivation or downregulation of the carbapenem-specific OprD porin, along with inducible AmpC production or efflux pump overexpression (MexAB-OprM), as significant contributors to imipenem resistance and reduced meropenem susceptibility in *P. aeruginosa* [31,32]. These findings underscore the complex interplay of different resistance mechanisms in the development of antibiotic resistance, with carbapenems resistance serving as a prominent example [29,33]. Furthermore, the concomitant upregulation of AmpC in conjunction with the inactivation of OprD contributes synergistically to the development of MDR against all commonly used antipseudomonal β-lactam antibiotics [34]. 

Efflux pumps and mutations in DNA gyrase enzymes (GyrA and GyrB) and type IV topoisomerases (ParC and ParE) both contribute to the development of FQ resistance in *P. aeruginosa*, enabling them to work in combination against these antibiotics [35]. Emerging research highlights *fusA1* gene mutations as an additional contributor to aminoglycoside resistance in *P. aeruginosa* [36,37]. These mutations, observed in clinical strains and particularly prevalent among CF patients, coincide with overexpression of MexXY efflux pumps [37,38,39].

By examining the diverse molecular mechanisms contributing to MDR in *P. aeruginosa*, it becomes evident that a combination of synergistic factors enables the effective protection of bacteria against antibiotics [40,41,42,43]. Notably, the active efflux of antibacterial agents through multidrug efflux pumps, in conjunction with the outer membrane (OM) permeability barrier, plays a pivotal role in defending against antibiotic threats [44,45]. This collaborative interplay between active efflux and the robust OM barrier gives rise to a synergistic MDR phenotype [43]. Understanding these intricate resistance mechanisms is essential for formulating targeted approaches to combat MDR and enhance treatment outcomes in *P. aeruginosa* infections. Building upon this foundation, the subsequent sections of this paper delve deeper into the specific mechanisms involved and explore potential strategies to overcome MDR challenges [46,47,48,49].

## 2. RND Efflux Pumps in *P. aeruginosa*, the Typhons of Antibiotic Resistance

Among the diverse enzymatic and mutational resistance mechanisms, multidrug efflux systems, particularly those categorized within the resistance-nodulation-cell division (RND) superfamily of drug/proton antiporters, are the main players associated with both intrinsic and acquired MDR, specifically in Gram-negative bacteria such as *P. aeruginosa* [50]. RND pumps act at the forefront of bacterial defense and reinforce the development of additional resistance mechanisms, allowing the emergence of MDR, XDR, and even pandrug-resistant (PDR) bacteria that show resistance to all antibacterial classes [51]. Efflux pumps are capable of expelling multiple unrelated antibiotic classes into its external environment, preventing their accumulation to toxic levels within bacterial cells and affording bacterial survival at higher antibiotic concentrations to promote MDR [52]. Efflux pumps of the RND family are versatile transporters capable of exporting a wide range of compounds, such as antibiotics, detergents, dyes, organic solvents, aromatic hydrocarbons, cationic biocides, lipophilic drugs, free fatty acids, amphiphilic agents, antiseptics, and compounds with anti-virulence and anticancer properties [53,54,55,56,57,58,59,60]. Interestingly, these molecules exhibit no shared chemical characteristics, except for their amphiphilicity. This unique feature, shared by antibiotics and cellular toxins, plays a vital role in their ability to move through fluids and target specific sites. Moreover, it enables these compounds to penetrate bacterial cells by crossing the lipid bilayer of the cell membrane [61,62]. Nonetheless, RND exporters display selectivity in their transport function, avoiding non-specific binding, as they do not facilitate the export of essential nutrients or non-toxic metabolites such as carbohydrates (e.g., glucose and fructose), amino acids, nucleotides, vitamins, and minerals. It is noteworthy that while the complete understanding of the physiological characteristics and endogenous substrates of RND efflux pumps remains limited, these proteins are known to have additional physiological functions beyond their role in MDR [63,64]. 

RND pumps not only drive the efflux of intrinsic toxic metabolites, safeguarding the bacterial milieu from their detrimental effects [65,66], but also play a role in modulating quorum sensing [67,68], influencing biofilm formation [69] and impacting bacterial virulence [70,71]. Thus, these versatile pumps contribute to fundamental cellular defense mechanisms [71,72]. Moreover, these proteins transport a specific range of antimicrobials. For instance, while the MexB efflux pump in *P. aeruginosa* and the AcrB efflux pump in *E. coli* do not possess the ability to extrude aminoglycoside antibiotics such as gentamicin and tobramycin, the MexY efflux pump in *P. aeruginosa* and the AcrD efflux pumps in *E. coli* are capable of effectively expelling them [61,73,74]. 

*Pseudomonas aeruginosa* possesses various efflux pump families linked to MDR (Figure 1 and Table 1), with the RND efflux pumps emerging as the most vital. RND pumps play a pivotal role in defending against antibiotics due to their unique structure and function [75,76,77]. These pumps efficiently expel a wide range of antimicrobial agents across the inner membrane (IM) and OM, reducing intracellular drug concentrations and conferring resistance. Additionally, RND pumps possess broad substrate specificity, enabling resistance to multiple antibiotic classes simultaneously. Their importance in *P. aeruginosa*’s MDR is evident from their association with clinical isolates exhibiting high-level resistance to multiple drugs [78,79].

Efflux pumps and their role in antibiotic resistance were first investigated in the 1980s when plasmid-transferred energy-dependent efflux pumps were found to contribute to tetracycline resistance in *E. coli* [117]. McMurry et al. (1980) revealed the capacity of efflux pumps to transfer drugs even against a concentration gradient, highlighting their role in MDR [117]. This discovery demonstrated a close connection between efflux pumps, OM permeability, and the development of MDR. Efflux systems, found in both pathogenic and non-pathogenic bacteria, have been unveiled prior to widespread exposure to antimicrobial agents, implying their intrinsic evolution as a resistance mechanism independent of antimicrobial use [53,118,119]. These systems empower bacteria to survive at suboptimal antibiotic concentrations, providing a window of opportunity for the development of sophisticated resistance mechanisms [120]. 

Efflux pumps play a crucial role in bacterial evolution and environmental adaptation due to several key factors [121,122,123]. Primarily, they can extrude a diverse range of potentially harmful substances, such as antibiotics, heavy metals, and host-derived antimicrobial peptides, which promotes bacterial survival in unfavorable conditions and leads to the development of multidrug resistance [124]. They also regulate the expression of numerous genes involved in critical bacterial processes, influencing virulence factors that enable bacteria to cause infections [125]. Furthermore, efflux pumps contribute to biofilm formation, facilitating bacterial adherence to various surfaces and providing protective structures that enhance survival and resistance to antibiotics [126,127]. Another critical aspect of their function is the regulation of quorum sensing, a complex communication system linked to population density that allows bacteria to coordinate group behaviors and adapt to changes in their environmental community [128,129]. Lastly, efflux pumps enable bacteria to mitigate a wide range of environmental stressors, aiding adaptation to diverse ecological niches and promoting evolutionary success [130]. Through these multifaceted contributions to bacterial survival strategies, efflux pumps play an indispensable role in bacterial evolution and ecology [131]. These pumps, widely distributed in nature, enabled bacteria to thrive in their ecological niches, protecting them from various harmful substances such as toxic compounds, antimicrobial molecules, reactive oxygen species (ROS), and degradation by-products [53,57,78,132,133,134]. RND efflux pumps exhibit a diverse range of substrate specificity, ranging from high selectivity for single compounds to the ability to transport a wide array of diverse compounds [54,78,135]. The recognition of specific substrates by these pumps is governed by the complex interplay between the physicochemical properties of the substrate, the binding site of the pump, and the surrounding environment, involving intricate mechanisms of molecular recognition and binding [43]. 

In *P. aeruginosa,* a comprehensive investigation has identified a diverse array of RND family efflux pumps, totaling twelve in number. These efflux pumps, including multidrug efflux AB-OM porin M (MexAB-OprM), MexXY-OprM, MexEF-OprN, and MexCD-OprJ (Figure 2), have been extensively studied for their notable contributions to MDR. Additionally, these efflux pumps possess distinct resistance mechanisms that have the potential to give rise to highly resistant strains, potentially even leading to the emergence of XDR or PDR phenotypes [1,51,136].

Among the three Mex pumps, MexAB-OprM exhibits greater prominence due to its wide substrate specificity, which contributes to MDR in *P. aeruginosa*, presenting a significant clinical challenge [60]. The upregulation of MexCD-OprJ is closely linked to elevated resistance in the majority of clinical strains, notably against ciprofloxacin, cefepime, and chloramphenicol, among other antibiotics [138]. In the presence of quinolones, MexEF-OprN is overproduced [139]. Moreover, the efflux activity of MexXY-OprM plays a pivotal role in conferring resistance to aminoglycosides, FQs, and zwitterionic cephalosporins in *P. aeruginosa* [140].

The RND efflux pumps form intricate tripartite protein complexes, functioning as drug/proton antiporters that facilitate the efficient expulsion of specific substrates across the outer membrane from the periplasmic space [141]. These pumps consist of three essential protein units: the inner-membrane protein (IMP) represented by MexB, MexY, MexD, or MexF; the periplasmic membrane fusion protein (MFP), also known as periplasmic adaptor proteins (PAPs), encompassing MexA, MexX, MexC, or MexE; and the outer membrane protein (OMP) encompassing OprM, OprJ, or OprN. Working in synergy, these components assemble into a trans-periplasmic channel, enabling the effective binding of substrates within the periplasm or cytoplasm [53] (Figure 2).

Upon the assembly of the protein complex, the RND efflux pump system progresses through distinct stages, namely the resting and transport stages [142,143]. Driven by the proton motive force (PMF) [144], the RND efflux mechanism undergoes a coordinated sequence of events [145,146], encompassing site access, substrate binding, and extrusion, facilitated by a precisely orchestrated rotational mechanism [61,116,145,147]. This intricate process empowers the RND system to operate as a highly efficient and dynamic pumping apparatus. Notably, in clinical isolates of *P. aeruginosa*, significant expression of RND pumps, specifically MexAB-OprM, MexCD-OprJ, MexEF-OprN, and MexXY-OprM, has been well documented and is further expounded upon in the subsequent section [148].

### 2.1. MexAB-OprM

The *MexAB-OprM* operon, the first characterized multidrug efflux pump in *P. aeruginosa* [95], plays a significant role in MDR [95,149,150,151]. As drug concentrations near the pump increase, MexB undergoes a conformational change, facilitating active expulsion of molecules through a periplasmic and OM tunnel formed by MexA and OprM [152]. Mutant strains lacking these efflux transporters display heightened sensitivity to multiple antibiotics, emphasizing the crucial role of constitutive expression of intrinsic efflux pumps in *P. aeruginosa* innate drug resistance [118].

The expression of this efflux pump is constitutive in wild-type *P. aeruginosa* strains and primarily regulated by repressor genes, namely, *mexR* [153], *nalC* [154], and *nalD* [155] (Figure 3). Selective pressure from antibiotic usage has favored *P. aeruginosa* strains with enhanced MexAB-OprM expression [156,157]. Mutations in the repressor genes have been linked to increased pump expression [158], particularly those causing translational disruptions, such as nonsense substitutions, frameshifts [157,159], and insertions by insertion sequences [158,160]. Additionally, non-synonymous substitutions that alter the repressors’ molecular structure have been implicated [156,161]. However, certain non-identical substitutions, such as V126E in *mexR*, have shown no correlation with increased efflux pump expression [162,163]. 

The MexAB-OprM efflux pump exhibits an extensive substrate range, effluxing a diverse array of antibiotics critical to *P. aeruginosa* therapy. It actively extrudes FQs, β-lactams, macrolides, tetracyclines, lincomycin, chloramphenicol, novobiocin, and others [96] (Table 1). This efflux pump’s significance extends beyond antibiotic resistance, as it also plays a key role in counteracting antimicrobial compounds derived from plants, exemplified by its ability to expel carvacrol and numerous other plant-based agents [98]. Carbapenemase-producing carbapenem-resistant *P. aeruginosa* strains often exhibit heightened expression of MexAB-OprM, contributing significantly to their resistance against carbapenems [164]. Furthermore, the simultaneous overexpression of MexAB-OprM and AmpC β-lactamase synergistically enhances *P. aeruginosa*’s resistance to a wide range of antipseudomonal β-lactams, exemplifying the intricate interplay between these mechanisms [165]. These discoveries unveil the vital role and exceptional adaptability of the MexAB-OprM efflux pump in the pathogen MDR, offering a unique perspective within the realm of antimicrobial research.

### 2.2. MexXY

The MexXY efflux pump is a distinct MDR efflux pump operon expressed by *P. aeruginosa*, notable for its absence of an OMF coding sequence. However, its functional complexity emerges as it collaborates with OprM from the *mexAB-oprM* operon to form a multidrug efflux pump [111]. Intriguingly, certain strains, including *P. aeruginosa* PA7 [166], feature an additional coding sequence within the *MexXY* operon for OprA, an OMF that can interact with MexXY [112,167]. This intriguing observation accentuates the intricate nature of the MexXY efflux pump, highlighting its pivotal role within the intricate landscape of AMR mechanisms in *P. aeruginosa*.

The MexXY efflux pump’s expression is induced by antimicrobial agents and regulated through several key repressors, including *mexZ* [168], the *parRS* two-component regulatory system [169], and the aminoglycoside-inducible anti-repressor *armZ* [170,171] (Figure 3). Notably, mutations in *mexZ* have been associated with increased MexXY expression, particularly observed in CF isolates [140,163,172]. Similar observations have also been made for *parR* and *parS* [38,169,173]. MexXY is renowned for its pivotal role in aminoglycoside resistance [174], frequently observed in strains harboring aminoglycoside-modifying enzymes (AMEs). The concerted action of MexXY and AMEs synergistically contributes to heightened aminoglycoside resistance [172].

The MexXY efflux pump, exhibiting a substrate profile akin to MexAB (Table 1), plays a crucial role in conferring broad-spectrum resistance against a diverse array of antipseudomonal antibiotics. Its impact on antibiotic resistance is particularly noteworthy in *P. aeruginosa* CF lung isolates, where MexAB functionality is compromised [175].

In the presence of OprA, the MexXY system expressed by the OprA-carrying PA7 strain significantly contributes to resistance against carbenicillin and sulbenicillin [112]. Moreover, specific amino acid substitutions in Mex efflux pumps, such as the F1018L mutation in the MexY component of the MexXY-OprM system, have been identified as key modulators of antibiotic resistance in *P. aeruginosa* [175]. This F1018L substitution, for instance, has been shown to double the bacterium’s resistance against aminoglycosides, cefepime, and FQs, highlighting the critical role of Mex pump structure in determining resistance profiles [175].

### 2.3. MexCD-OprJ

The *mexCD-oprJ* operon in *P. aeruginosa* typically remains silent or expressed at low levels under optimal or standard laboratory conditions [176]. As a result, it does not significantly contribute to the innate antibiotic resistance of the bacterium. However, when overexpressed, it is associated with resistance to multiple antibiotic classes [99,138]. The expression of MexCD-OprJ is primarily regulated by the repressor gene nfxB, which is located adjacent to the *mexCD-oprJ* operon, and it is transcribed in the opposite direction (divergently) to this operon. This arrangement allows *nfxB* to control the expression of MexCD-OprJ by binding to specific regulatory regions in the DNA [99] (Figure 3). Mutations in *nfxB*, including nucleotide deletions, missense, and nonsense mutations, are linked to increased MexCD-OprJ expression [138,177].

MexCD-OprJ is renowned for its role in conferring resistance to FQs, such as levofloxacin and ciprofloxacin [97,116,176]. It also exhibits efflux activity against a diverse array of antimicrobial agents, including macrolides, novobiocin [96], tetracyclines, chloramphenicol [138,178,179], zwitterionic cephalosporins (cefepime and cefpirome) [180], and the biocide chlorhexidine [100] (Table 1). Mutations in the *mexD* gene have been associated with alterations in MexCD-OprJ’s substrate specificity, leading to resistance against carbenicillin [181], ceftolozane-tazobactam, and ceftazidime-avibactam [177]. More recently, Sanz-García et al. (2022) reported that subinhibitory concentrations of ciprofloxacin can drive the emergence of *P. aeruginosa* mutants with elevated MexCD expression levels, resulting in cross-resistance to antibiotics from distinct structural families, such as ceftazidime, amikacin, aztreonam, imipenem, and fosfomycin [131].

Variants of the *mexCD-oprJ* genes have been identified in mobile genetic elements not only in *P. aeruginosa* but also in various other bacterial species [101,182]. These mobile efflux pump operons, known as *tmexCD-toprJ* (where “*t*” denotes transferable), are believed to have originated from *Pseudomonas* spp. chromosomes [102]. Extensive research has revealed multiple distinct *tmexCD-toprJ* variants, found in either the bacterial chromosome or plasmids, and their presence has been directly linked to tetracycline resistance as well as reduced susceptibility to diverse classes of antibiotics under controlled laboratory conditions [101,102].

### 2.4. MexEF-OprN

Similar to MexCD-OprJ, the MexEF-OprN system remains mostly dormant in wild-type strains cultured under standard laboratory conditions. However, when this system is activated and overexpressed, it confers resistance to a range of antibiotics, including chloramphenicol, FQs, and trimethoprim [103,183] (Table 1). The regulatory mechanisms governing the MexEF-OprN efflux pump involve two key elements: *mexT*, a transcriptional activator with *lysR*-like characteristics, and *mexS*, an oxidoreductase enzyme with an elusive enzymatic role [184,185]. Notably, these regulatory genes are located upstream of the *mexEF-oprN* operon [168] (Figure 3). In wild-type *P. aeruginosa* strains, inactivating mutations in the *mexT* gene are frequently observed, resulting in the upregulation of MexEF-OprN expression [184]. Furthermore, *mexS* expression is influenced by *mexT* [184], and multiple mutations in the *mexS* gene, predominantly detected in clinical strains, are believed to contribute to the enhanced activity of MexEF-OprN [185,186,187,188].

The upregulation of MexEF-OprN has been found to coincide with the downregulation of the porin OprD, which is thought to be mediated by the dual role of the MexEF-OprN activator MexT as a repressor of *oprD* [184,189]. Inactivating mutations and reduced expression of *oprD* have been linked to resistance against carbapenems [158] and colistin [190]. This observation provides a plausible explanation for the observed resistance of MexEF-OprN overexpressing strains to antibiotics that are not typically substrates of this efflux pump and are not efficiently expelled by it [190].

**Figure 3 antibiotics-12-01304-f003:**
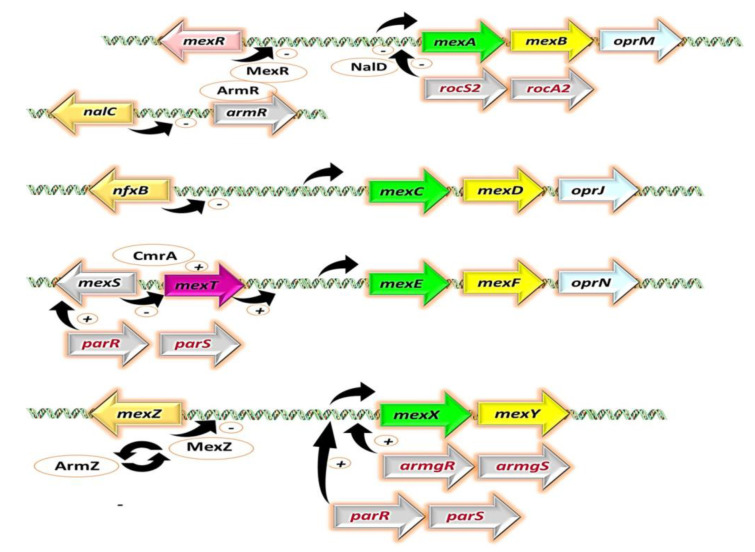
Regulation systems controlling the Mex efflux pumps in *P. aeruginosa*. Genes are represented by arrows, while proteins are depicted as oval shapes. The membrane fusion proteins (MFPs) are shown in green, the RND proteins or IMP in yellow, and the outer membrane factors (OMFs) in pale cyan. Repression is indicated by a “-” sign, and activation is represented by a “+” sign. Adapted with permission from ref. [168], 2018, Issa et al. Licensed under Creative Commons Attribution 4.0 International License. This figure was partly generated using Servier Medical Art, provided by Servier, licensed under a Creative Commons Attribution license (CC-BY).

## 3. The Perilous Dance: Unveiling the Clinical Relevance of Mex Efflux Pumps in *P. aeruginosa* Isolates

In clinical isolates, the concurrent mutational overexpression of specific major efflux pumps, such as MexAB-OprM combined with MexCD-OprJ or MexEF-OprN paired with MexXY, has been identified as a significant contributor to mutation-driven MDR [22,29,191,192]. Several studies have consistently indicated that the overexpression of MexAB-OprM and MexXY is frequently observed (ranging from 10% to 30%) in clinical isolates [33,97,139,158,176,193,194,195], highlighting their significant role in antibiotic resistance. Conversely, the occurrence of MexCD-OprJ and MexEF-OprN overexpression, as suggested by existing research, is comparatively uncommon, with a prevalence of less than 5% among clinical isolates [33,139,158,196,197,198]. Owing to its prominent role, the MexAB-OprM efflux pump occupies a significant position among the Mex pump family in clinical isolates, being the foremost contributor to MDR. It confers reduced susceptibility to a wide range of existing antipseudomonal antibiotics, encompassing nearly all classes, with the exception of imipenem [199].

In a recent study by Rahbar et al. (2021) [97], Mex efflux pump expression was analyzed in 122 clinical *P. aeruginosa* isolates. MexY emerged as the prominent pump (expressed in 74.6%), followed by MexB (69%). MexE and MexC showed lower expression rates of 43.4% and 28.7%, respectively. The study revealed a significant correlation between Mex pump expression and resistance to antipseudomonal antibiotics. Ticarcillin exhibited the highest resistance (80%), followed by ciprofloxacin (74%) and meropenem (71%). The presence of an efflux pump inhibitor (EPI), phenylalanine-arginine β-naphthylamide (PAβN), enhanced susceptibility to imipenem, piperacillin, gentamicin, ciprofloxacin, and levofloxacin. These findings underscore the clinical relevance of Mex pumps, especially MexY and MexB, in MDR development in *P. aeruginosa* isolates.

In a notable investigation led by Horna et al. (2018), the interplay between MexEF-OprN and MexAB-OprM was explored within a clinical population. Examining 190 clinical isolates of *P. aeruginosa* collected from patients in Peru (December 2012–June 2013), the study uncovered intriguing findings. Among these isolates, 90 exhibited efflux pump overexpression, with one-third showing relevant modifications in OprD linked to carbapenem resistance [158]. Similarly, another third displayed modifications in *nalC*, *nalD*, or *mexR*, associated with MDR, reduced carbapenem susceptibility, OprD alterations, and extensive biofilm production. Co-analysis of all regulators revealed a significant association between relevant modifications in MexAB-OprM regulators and MDR, specifically when the *mexS* gene remained intact to negatively regulate MexEF-OprN expression. This emphasizes the crucial role of the intact *mexS* gene in the complex dynamics between MexAB-OprM and MexEF-OprN [158].

In a study conducted by Goli et al. (2016) in Tabriz, Iran, 100 clinical isolates of *P. aeruginosa* were analyzed [193]. The findings revealed a high prevalence of MDR in these isolates, with 71% exhibiting resistance to multiple antibiotics from different classes such as FQs, β-lactams, and aminoglycosides. The study also investigated the role of efflux pump genes in MDR phenotype. Moreover, the overexpression of efflux pump genes, specifically *mexB* (62%) and *mexY* (65%), indicated their prominent and significant involvement in MDR. Furthermore, the study examined the impact of the EPI PAβN on antibiotic susceptibility. Among the isolates resistant to ciprofloxacin and levofloxacin, PAβN was able to significantly reduce the MIC in 39.6% and 28.5% of cases, respectively [193]. 

Shigemura et al. (2015) examined the connection between the expression of multidrug efflux pump genes and MDR in 105 clinical strains of *P. aeruginosa* isolated from urinary tract infections (UTIs) in patients treated at three hospitals in Hyogo prefecture, Japan [176]. Their findings demonstrated a significant correlation between elevated expression of *mexC* or *mexB* and the occurrence of complicated UTIs. Moreover, they identified a specific connection between increased *mexC* expression and resistance to the antibiotic levofloxacin. Thus, their findings demonstrated the simultaneous involvement of MexB and MexC in FQ resistance [176].

Terzi et al. (2014) conducted a comprehensive investigation of 50 *P. aeruginosa* clinical isolates sourced from diverse laboratories in Turkey during the period of November 2010 to November 2011 [198]. These isolates were meticulously classified into distinct resistance profiles, encompassing the MDR, carbapenem-resistant (CR), quinolone-resistant (QR), and carbapenem and quinolone-resistant (CQR) groups. The analysis of these isolates revealed significant occurrences of gene overexpression, particularly observing high frequencies of overexpressed *mexD*, *mexB*, *mexF*, and *mexY* genes, with percentages of 88%, 76%, 46%, and 40%, respectively. Remarkably, within the MDR group, *mexB*, *mexD*, *mexF*, and *mexY* genes demonstrated consistent patterns of overexpression across the majority of isolates. Additionally, the study provided compelling evidence of a direct correlation between the increased transcription of *mexB* and meropenem resistance, as well as the involvement of MexCD-OprJ and MexEF-OprN in FQ resistance. Notably, the MexCD-OprJ efflux pump was found to be associated with MDR, while the upregulation of *mexY* contributed to the development of gentamicin resistance [198]. Conversely, separate research conducted by Ozer et al. (2012) in Turkey failed to establish a significant correlation between *mexB*, *mexY* overexpression, and a certain group of antibiotics such as β-lactam drugs [197].

In another study conducted by Llanes et al. (2011) in France, the expression patterns of efflux pump genes were examined in 85 non-CF strains of *P. aeruginosa* [139]. The study focused on the overexpression of *mexB* and *mexY*, which were found in 36.4% and 45.8% of the strains, respectively. Additionally, the MexEF-OprN efflux pump was observed to be upregulated in 10 clinical isolates, alongside the upregulation of MexAB-OprM and MexXY/OprM. Further analysis revealed the presence of mutations in the *mexT*, *mexS*, and *mvaT* genes among the MexEF-OprN overproducers, known as *nfxC* mutants. Notably, ciprofloxacin exposure selectively favored the growth of *nfxC* mutants, indicating the predominant role of MexEF-OprN in conferring resistance to FQs. These findings suggest the potential prevalence of *nfxC* mutants in clinical settings and highlight the need for further exploration of the regulatory mechanisms underlying this efflux system [139].

In a Brazil-based study conducted by Xavier et al. (2010), the intricate landscape of MDR mechanisms was explored in 59 clinical isolates of *P. aeruginosa* [196]. These isolates were derived from patients with bloodstream infections at a São Paulo hospital. Notably, the study revealed varying degrees of overexpression, with *mexB* exhibiting upregulation in approximately one-quarter (27.1%) of the isolates, while *mexY* showed elevated expression in slightly more than half (50.8%). In contrast, the MexEF-OprN and MexCD-OprJ systems displayed negligible signs of overexpression, deviating from previous studies, possibly influenced by the unique infection location. Additionally, the study uncovered AmpC β-lactamase overexpression in around one-tenth (11.9%) of the *P. aeruginosa* isolates. Furthermore, a significant decrease in *oprD* expression was observed in nearly three-quarters (69.5%) of the isolates and in the majority (87.1%) of imipenem non-susceptible strains. These findings shed light on the intricate interplay between efflux systems, AmpC β-lactamase, and oprD downregulation, underscoring their significant role in MDR among clinical isolates of *P. aeruginosa* [196].

Overall, Mex pumps wield indispensable significance within clinical settings and isolated *P. aeruginosa* strains. By actively extruding antibiotics, these efflux pumps fuel the development of resistance, thwarting the efficacy of conventional therapeutic agents. A comprehensive understanding of the presence, dynamics, and interplay of Mex pumps is imperative for devising innovative interventions to surmount the formidable challenges posed by MDR, elevating the prospects of successful treatment outcomes in clinical practice.

## 4. Athena’s Wisdom: Non-Antibiotic Strategies against *P. aeruginosa* Efflux Pumps

### 4.1. Achilles’ Heel Exposed: EPIs Target P. aeruginosa MDR

With their well-established role in driving *P. aeruginosa* MDR and pathogenicity and the limited availability of new antibiotics to counter resistance, the efflux pumps have emerged as paramount targets for innovative drug development aiming to surmount multidrug resistance [200]. An emerging therapeutic approach entails the utilization of compounds that exhibit EPI properties, offering promising prospects for MDR [201]. EPIs can prevent the emergence of mutant resistance by targeting and inhibiting the efflux pumps responsible for drug resistance in bacteria [136]. When EPIs are used in combination with antibiotics, they block the activity of efflux pumps, preventing them from pumping out the antibiotics from the bacterial cells. This effectively keeps the antibiotic concentration inside the cells at a level sufficient to inhibit bacterial growth and prevent the emergence of mutant strains that are resistant to the antibiotics. By inhibiting the efflux pumps, EPIs help to maintain the efficacy of antibiotics and reduce the development of resistance [200]. In addition to preventing the emergence of mutant resistance, EPIs offer the additional benefit of reducing biofilm formation and antibacterial tolerance within biofilms [202]. By inhibiting efflux pumps, EPIs disrupt the secretion of signalling molecules necessary for the establishment and maintenance of biofilms [123,127]. Therefore, the attachment and growth of bacteria within the biofilm are adversely affected by EPIs, leading to a less structured and robust biofilm [203,204]. This is particularly important because robust biofilms are often essential for persistent bacterial infections [205,206]. As EPIs interfere with biofilm formation and maintenance, they also contribute to the reduction in antibacterial tolerance within biofilms [202,207]. Biofilms often serve as a protective barrier against antibiotics, but the disruption of biofilm structure by EPIs increases the susceptibility of bacteria to antibacterial agents [123,202,204,208]. Furthermore, this inhibition prevents the efflux of intracellular substances, including quorum sensing molecules that are essential for biofilm development [123,209,210]. EPIs can also halt the secretion of virulence factors outside the bacterial cells, which are crucial for the pathogenicity of bacteria. This dual action of EPIs in reducing biofilm formation and inhibiting virulence factor secretion contributes to their potential as valuable tools in combating bacterial infections [211,212]. EPIs, when employed as adjuvants in combination with antibiotics, offer a promising approach to revive the activity of antibiotics that have already become ineffective, thereby restoring their diminished antimicrobial potencies [123,213]. As an example, PAβN (Figure 4), a well-researched and extensively developed EPI of Mex pumps, has garnered significant attention [123,214,215,216].The application of PAβN results in a noteworthy reduction in the inherent susceptibility of *P. aeruginosa* to fluoroquinolone antibiotics, such as levofloxacin, with a substantial 8-fold increase in susceptibility. Moreover, the development of acquired resistance arising from efflux pump overexpression is significantly impeded, as evidenced by a remarkable 32- to 64-fold reduction in the minimum inhibitory concentration (MIC) of levofloxacin. Additionally, strains with heightened efflux pump expression and specific target mutations conferring resistance to levofloxacin (e.g., gyrA and parC) exhibit notable 32- to 64-fold MIC reductions in the presence of PAβN [215]. 

Despite extensive research and dedicated efforts over the course of the last two decades, the clinical approval of EPIs, especially small molecule inhibitors, remains elusive [147,217]. The clinical approval of EPIs continues to pose a significant challenge, primarily attributed to the intricate balance between their required activity concentrations and observed toxicity effects, as well as their suboptimal pharmacokinetic properties and limited in vivo efficacies [43,50]. To date, MP-601,205 (Figure 5) stands as the sole EPI subjected to extensive clinical trials, signifying significant progress in EPI therapy development [123,218,219]. This bis(pyrimidine)sulfonamide compound, along with the antibiotic ciprofloxacin, was explored as a potential therapeutic strategy for respiratory infections in individuals with CF and ventilator-associated pneumonia caused by *P. aeruginosa*, utilizing an inhalable aerosol formulation. [95,220]. Development of the compound was halted during phase I clinical trials due to concerns regarding tolerability and cytotoxicity in mammalian cells [123,218,219]. There is currently no EPI under clinical investigation or approved for clinical use. The toxicity issues associated with EPIs, which require high doses for significant activity, have contributed to the absence of ongoing trials [221].

The development of an ideal EPI for Gram-negative bacteria, such as *P. aeruginosa*, is guided by several widely accepted criteria. Firstly, an effective EPI should lack direct antibacterial activity, with an MIC above 100 μM, as the viability of bacteria does not rely on active efflux pumps [148]. Moreover, it should not interact with secondary non-specific targets such as the inner (IM) or OM [221]. In addition to these criteria, the ideal EPI must overcome the OM barrier to reach its target, thereby enhancing the activity of antibiotics that are substrates of efflux pumps. It should also exhibit no activity against mutants lacking the respective efflux pumps, demonstrating its direct targeting of efflux pumps [53]. Furthermore, the EPI should reduce extrusion and promote the accumulation of efflux pump substrates [222,223]. To be considered suitable for clinical use, the EPI should not disrupt the proton gradient across the cytoplasmic membrane, as it could adversely affect the energy dynamics of both bacterial and eukaryotic cells [54,55]. Moreover, it is imperative that the EPI exhibits advantageous pharmacological attributes, such as minimal toxicity, notable therapeutic efficacy, and optimal safety measures, alongside optimal ADMET (Absorption, Distribution, Metabolism, Excretion, and Toxicity) characteristics. Furthermore, the EPI should demonstrate resilience in serum, ensuring its stability and effectiveness, while also presenting economic viability for large-scale manufacturing [200,224]. By meeting these criteria, the pursuit of an ideal EPI shows great potential in tackling the obstacles presented by efflux pumps in Gram-negative bacteria. This opens up fresh opportunities for combating MDR and augmenting the effectiveness of antibiotic treatment. 

While several compounds have exhibited promising potential as EPIs against *P. aeruginosa*, their precise mechanisms of action (MoA) remain largely uncharacterized. Consequently, it is challenging to classify these molecules based on their specific mode of action. To address this knowledge gap, an alternative classification system based on the source of EPIs has been proposed. This approach categorizes EPIs into three broad groups: phytochemicals (derived from plant products), synthetic or semisynthetic compounds, and EPIs derived from microorganisms [200]. In the context of combating *P. aeruginosa*, this section focuses on elucidating the EPIs within these categories that specifically target the Mex efflux pumps.

#### 4.1.1. Plant-Derived EPIs

Plant-derived phytochemicals have gained attention for their ability to enhance the efficacy of antibiotics by several times through synergistic interactions against bacteria [225,226,227,228,229,230,231,232,233,234,235,236,237,238].

These compounds exhibit a dual action, as they possess inherent antibacterial properties while also acting as EPIs. For example, *Berberis vulgaris* produces berberine, an antibiotic, along with 5′-methoxyhydnocarpin (5′-MHC), an EPI [239,240,241]. *Lupinus argenteus* contains α-linolenic acid, an antibiotic, and isoflavone-based EPIs. Additionally, *Dalea versicolor* synthesizes dalversinol A, an antibiotic, and EPI methoxychalcone [239,240,241]. Plants have evolved mechanisms to protect themselves against bacterial invasion, including the expression of efflux pumps by bacterial pathogens [239,242]. In turn, plants have developed EPI compounds as a sophisticated delivery system to enable the entry of their antimicrobial phytochemicals into bacterial cells [239,243,244]. This is achieved through the production of phytoanticipins, which are constitutively present, and phytoalexins, whose levels dramatically increase upon microbial invasion [245,246,247,248,249]. Notably, plants lacking specific phytoalexins have been found to be more susceptible to microbial pathogens, highlighting the potential of these molecules as potent EPIs [245,246]. Among the various classes of phytoalexins, alkaloids, glyco-steroids, and terpenoids, alkaloids possess favorable chemical structures for serving as EPIs. Effective EPIs often feature multiple aromatic structures, at least one of which contains a nitrogen atom—a characteristic commonly found in alkaloids [250]. The chemical diversity and synergistic effects of plant-derived phytochemicals and their dual action as both antibiotics and EPIs make them promising candidates for combating *P. aeruginosa* efflux pumps and addressing MDR. 

Studies have provided compelling evidence that plant-derived alkaloids possess remarkable EPI properties, synergistically enhancing the efficacy of various antipseudomonal drugs. Notably, the *Apocynaceae*, *Berberidaceae*, *Convolvulaceae*, *Cucurbitaceae*, *Fabaceae*, *Lamiaceae*, and *Zingiberaceae* plant families are recognized as excellent sources of EPIs [239]. It is noteworthy that most plant-derived EPIs exhibit higher activity against efflux pumps in Gram-positive bacteria compared to Gram-negative strains. The complex tripartite pump arrangement alongside OM poses a challenge in discovering potent EPIs effective against Gram-negative bacteria [239]. However, several phytochemicals have demonstrated EPI characteristics against highly resistant Gram-negative bacteria, including *P. aeruginosa* [227,239].

Outlined below are two noteworthy subclasses of plant-derived EPIs that exhibit potent inhibition of Mex efflux pumps in *P. aeruginosa* [199,204].

Alkaloids**Conessine** (Figure 6), a steroidal alkaloid derived from *Holarrhena antidysenterica*, has been studied for its effects on *P. aeruginosa* Mex pumps. Research conducted by Siriyong et al. in 2017 demonstrated that conessine can significantly reduce the MICs of various antipseudomonal antibiotics such as levofloxacin, cefotaxime, and tetracycline by at least 8-fold in a strain overexpressing the MexAB-OprM efflux pump. Interestingly, conessine does not appear to act as a membrane permeabilizer, as it does not cause the accumulation of 1-N-phenylnaphthylamine [251].**Catharanthine** (Figure 6), Dwivedi et al. (2018) evaluated catharanthine, a terpene indole alkaloid isolated from *Catharanthus roseus*, against MDR *P. aeruginosa* strains. They found that catharanthine exhibited no intrinsic antibacterial activity, but when combined with tetracycline and streptomycin, it significantly reduced their MICs, increasing *P. aeruginosa*’s susceptibility and reducing the emergence of tetracycline-resistant mutants. Furthermore, catharanthine showed efficacy comparable to EPIs such as PAβN and displayed activity against Mex pump-expressing *P. aeruginosa* strains [252].**Berberine** and **palmatine** (Figure 6), extracted from *Berberis Vulgaris*, exhibit EPI activity [253]. In a study by Aghayan et al. (2017) on *P. aeruginosa* isolates from burn infections, these compounds significantly reduced the MIC of antipseudomonal antibiotics, such as ciprofloxacin, by up to 8-fold in strains overexpressing MexAB-OprM [253]. Furthermore, Su et al. (2018) demonstrated the synergistic interaction of berberine with imipenem, effectively reversing imipenem resistance in *P. aeruginosa* by inhibiting the MexXY OprM efflux pump. These findings highlight the potential of berberine and palmatine as EPIs for combating MDR in *P. aeruginosa* [254].**Theobromine** (Figure 6), an alkaloid derived from *Theobroma cacao*, has been identified as an inhibitor of the *P. aeruginosa* MexAB-OprM efflux pump [239]. According to research by Piddock et al. (2010), it was suggested that compounds with small heterocyclic or nitrogen-containing structures have the potential to act as inhibitors of RND efflux pumps. In their study, a library of 26 compounds was evaluated, and among them, theobromine emerged as a highly potent plant-based EPI [255]. The compound demonstrated significant efficacy in reducing the MIC of ciprofloxacin in *P. aeruginosa* strains overexpressing the MexAB-OprM efflux pump. These findings support the notion that theobromine holds promise as an effective EPI in combating antibiotic resistance in *P. aeruginosa* [255].Phenolic compounds**p-Coumaric acid** (Figure 6) is a phenolic acid that can be extracted from a variety of edible plants such as *Gnetum cleistostachyum* [239,256]. In a recent study, Choudhury et al. (2016) investigated p-Coumaric acid and its derivative as a potential EPI of the MexAB-OprM in *P. aeruginosa*. These compounds showed promising results in preliminary screening, including significant docking scores and the ability to enhance the activity of ciprofloxacin in MexAB-OprM overexpressing strains of *P. aeruginosa*. The researchers suggest that these compounds could serve as lead molecules for the development of MexAB-OprM inhibitors, offering a potential solution to combat multidrug-resistant *P. aeruginosa* infections [256].**Curcumin** (Figure 6) is a polyphenol curcuminoid derived from the rhizomes of *Curcuma longa*. It has been investigated for its potential as an adjuvant to enhance the antimicrobial activities of commonly used antibiotics against MDR *P. aeruginosa*. In a study by Negi et al. (2019), curcumin (50 mg/L) was found to significantly decrease the MIC values of various antipseudomonal drugs, including meropenem, carbenicillin, ceftazidime, gentamicin, and ciprofloxacin, when used in combination with them against 170 clinical *P. aeruginosa* isolates. This synergistic effect suggests the promising role of curcumin in combating drug-resistant bacterial infections and its potential as an EPI [257].**Resveratrol** (Figure 6), a polyphenol stilbenoid found in various fruits and vegetables, including peanuts, blueberries, cranberries, and Japanese knotweed [258,259], has shown promising antimicrobial and antibiofilm properties when combined with colistin against colistin-resistant *P. aeruginosa*. Wang et al. (2023) demonstrated that resveratrol enhanced the activity of colistin against resistant *P. aeruginosa* in vitro and improved colistin efficacy in vivo [259]. However, further investigation is needed to evaluate the impact of resveratrol on Mex efflux pumps in *P. aeruginosa*.

In the search for EPIs with reduced cytotoxicity, several phytochemical candidates have been explored [225,226,227,228,229,230,231,232,233,234,235,236,237,238]. However, their structural complexity poses challenges for conducting structure–activity relationship (SAR) studies and hinders their further advancement [245,260,261,262]. Nonetheless, the chemical structure of phytochemicals can serve as a valuable model for the design of highly potent synthetic EPIs, offering new avenues for development [263].

#### 4.1.2. EPIs of Synthetic Origin

The screening of small molecule semi-synthetic or synthetic expanded chemical libraries is a convenient approach for the identification of effective EPIs. Such synthetic small molecule EPIs can be further classified as discussed below [200], offering valuable insights into their structural features, MoA, and potential applications in combating drug resistance.


**Peptidomimetic Compounds**


In 1999, Microcide Pharmaceuticals and Daiichi Pharmaceutical Company introduced PAβN, a peptidomimetic EPI targeting RND efflux pumps in Gram-negative bacteria [215]. This EPI effectively inhibited the growth of *P. aeruginosa* strains overexpressing major Mex efflux pumps when used in combination with subinhibitory concentrations of levofloxacin, an antibiotic substrate of RND efflux pumps. However, despite its notable efficacy, PAβN’s clinical potential is limited due to its toxicity in mammalian cells [218,264,265]. PAβN is structurally composed of two amino acid moieties: L-phenylalanine (aa1) and L-arginine (aa2) connected to a naphthylamine (Cap) [136,148] (Figure 7. Numerous studies have investigated the SAR of PAβN to minimize cytotoxicity and improve its potency and ADME (absorption, distribution, metabolism, and excretion) properties [265,266,267]. Around 500 analogues were synthesized and tested based on PAβN’s chemical structure [265,266,267]. These modifications were implemented to enhance the EPI properties of PAβN, targeting specific aspects:**Replacement of aa1 and aa2**: For the optimization of PAβN, it is crucial to consider the replacement of amino acids 1 and 2. These amino acids should possess both aromatic and basic properties, with the possibility of reversing their order [265,267]. Notably, substituting L-phenylalanine with L-homo-phenylalanine has been shown to significantly enhance EPI potency by two-fold. Furthermore, alternative basic amino side chains, such as ornithine or aminomethylproline, offer viable options for substitution, expanding the scope of EPI modifications [265,267] (Figure 7 and Figure 8).**Substitution of the amide bond:** Introducing methylation to the amide bond between amino acid 1 and amino acid 2 results in a modest enhancement in both potency and plasma stability compared to the original compound [265] (Figure 7 and Figure 8).**Modification of the cap moiety:** The replacement of the naphthyl moiety with alternative fused rings, such as 5-aminoindan and 6-aminoquinoline, leads to a reduction in PAβN toxicity while enhancing the pharmacological properties of PAβN and its derivatives [265]. Notably, the incorporation of a 3-aminoquinoline moiety is crucial for mitigating cytotoxicity in mammalian cells during in vitro experiments [265] (Figure 7). These extensive SAR investigations have yielded a range of derivatives, including MC-02,595 and MC-04,124 [267,268,269], which exhibit potent EPI activities (Figure 8).

Despite showing elevated basic EPI properties of reduced toxicity, enhanced stability, and improved solubility, none of the active analogues sufficiently decreased the problems associated with the parent molecule. Therefore, PAβN and its novel derivatives are now restricted in their use as laboratory standard compounds that can be used to determine the level of inhibitor-sensitive efflux for specific antibiotics in various bacterial pathogens [221].

2.
**Arylpiperidines and Arylpiperazine Derivatives**


Arylpiperazines were first introduced in 2005 by Bohnert and Kern, who showed that 1-(1-naphthylmethyl)-piperazine (NMP) (Figure 9) is one of the most active compounds of this class of molecules, exhibiting EPI activity that enhances the susceptibility to levofloxacin and ethidium bromide (EtBr) by acting on RND pumps in *E. coli* that overexpresses AcrAB and AcrEF [136,270].

In addition, Ferrer-Espada et al. (2019) demonstrated the EPI activity of NMP, which reduces multidrug resistance (MDR) to antibiotics such as levofloxacin and piperacillin in *P. aeruginosa* strains, including wild-type and MexAB-OprM overexpressing strains [271]. However, the effectiveness of NMP in reducing resistance was lower compared to PAβN [271]. It is important to consider the potential toxicity of arylpiperazines, including NMP, to mammalian cells due to their serotonin reuptake inhibitor properties [270].

3.
**Pyridopyrimidine and Pyranopyridine Derivatives**


In 2003, pyridopyrimidine analogues were discovered as potent compounds against the MexAB-OprM pump in *P. aeruginosa* [272]. Among these analogues, D13-9001 or ABI-PP ([[2-({[((3*R*)-1-{8-{[(4-*tert*-butyl-1,3-thiazol-2-yl)amino]carbonyl}-4-oxo-3-[(*E*)-2-(1*H*-tetrazol-5-yl)vinyl]-4*H*-pyrido[1,2-*a*]pyrimidin-2-yl}piperidin-3-yl)oxy]carbonyl}amino) ethyl](dimethyl) ammonio] acetate) (Figure 10) demonstrated strong efficacy as a MexAB-OprM-specific EPI in both in vivo and in vitro studies [272,273,274]. In a *P. aeruginosa* strain overexpressing MexAB-OprM, this compound also demonstrated a remarkable 8-fold reduction in the MIC of aztreonam and levofloxacin [274]. Additionally, in a rat model of acute pulmonary infection caused by *P. aeruginosa*, the combination of D13-9001 and aztreonam showed a markedly higher survival rate compared to aztreonam monotherapy [272,274]. X-ray crystallography data reveal that D13-9001 exerts its EPI activity through distinct mechanisms. It utilizes both hydrophilic and hydrophobic components to disrupt the normal functioning of the pump. The hydrophilic part interacts with the substrate binding channel, preventing substrate binding, while the hydrophobic moiety tightly binds to a hydrophobic trap within the deep substrate binding pocket. These complementary interactions collectively lead to a reduction in antibiotic efflux. This multifaceted approach demonstrates the potential of D13-9001 as an effective EPI in combating MDR *P. aeruginosa* [221,273].

The development of D13-9001 as an EPI was hampered by nephrotoxicity, which was linked to the positively charged amine groups necessary for penetrating *P. aeruginosa*’s OM [218]. Notably, EPIs such as pyridopyrimidine derivatives demonstrate selectivity towards specific MDR efflux pumps, such as AcrB and MexB, while being ineffective against MexY [274]. These findings underscore the significance of comprehending substrate recognition and inhibition mechanisms for the development of targeted EPIs aimed at effectively combating MDR [61].

The introduction of pyranopyridines as potent EPIs by MicroBiotix in 2014 has provided novel avenues for addressing MDR [275]. Among these, MBX2319 (Figure 11), a synthetic pyrazolopyridine compound, exhibited remarkable effectiveness in increasing the activity of antibiotics such as ciprofloxacin levofloxacin, and piperacillin, by up to 8-fold against *E. coli* strains overexpressing AcrAB-TolC efflux pump [276].

SAR studies have further optimized the compound’s properties and identified key regions for activity, guiding the development of more effective EPIs. Nguyen et al. (2015) synthesized 60 analogs of MBX2319, generating valuable insights to improve potency, metabolic stability, and solubility [277] (Figure 11).

MBX2319 also exhibits strong EPI activity against *P. aeruginosa*, especially when combined with polymyxin B nonapeptide, which disrupts the OM of the bacterium [278]. Thus, the efficacy of MBX2319 is hindered by its inability to penetrate the highly selective OM of *P. aeruginosa* [279]. This highlights the main challenge in achieving broad-spectrum activity of pyranopyridine EPIs, particularly in Gram-negative bacteria such as *P. aeruginosa*, where OM penetration is a critical factor [279]. According to an in-depth X-ray crystallography investigation, MBX2319 was found to engage with the hydrophobic trap of the AcrB pump using its pyridine ring, potentially establishing a ring stacking interaction with specific amino acid residues [221]. This detailed structural analysis offers valuable insight into the mechanism by which MBX2319 functions as an EPI [221].

4.
**TXA Compounds**


TXA09155 (Figure 12), an innovative EPI developed by Taxis Pharmaceuticals, shows promising potential for treating MDR *P. aeruginosa* infections in burn victims and CF patients [136]. It effectively inhibits MexAB-OprM and MexXY-OprM efflux pumps, leading to enhanced efficacy of important antibiotics such as FQs, tetracyclines, and β-lactams [280]. This remarkable EPI candidate, derived from extensive synthetic screening, represents a significant advancement in the pursuit of effective treatment options for *P. aeruginosa* infections. It is derived from TXA01182 and a class of compounds known as aryl-alkyl diaminopentanamide EPIs [136,280] (Figure 12). These compounds were intelligently engineered to prevent membrane disruption and mitigate toxicity concerns associated with non-specific binding. The primary objective was to enhance metabolic and serum stability, surpassing traditional peptidomimetic EPIs such as PAβN [136]. To optimize the design, diverse druggable fused heterocycles were introduced, namely, benzothiazole, benzimidazole, benzofurane, benzothiophene, indole, and azaindole. These were connected to a fluorobenzene and a chiral diamine, resulting in the development of TXA01182. The aim was to minimize membrane disruption, address toxicity concerns related to non-specific binding, and improve metabolic and serum stability compared to traditional peptidomimetic EPIs such as PAβN [281]. Moreover, to enhance the potency of the EPI, the diamine side chain was structurally constrained into a pyrrolidine ring, resulting in the creation of TXA09155 [280].

In their study, Zhang et al. (2022) investigated the impact of TXA09155 on *P. aeruginosa*. They discovered that TXA09155 had no detrimental effects on the bacterial membrane, PMF, or ATP levels at concentrations below 50 µg/mL [280]. Additionally, TXA09155 effectively enhanced the potency of cefpirome and levofloxacin by 8-fold against *P. aeruginosa* strains overexpressing MexAB-OprM or MexXY-OprM. These findings were further confirmed by using MDR *P. aeruginosa* clinical isolates. Notably, TXA09155 demonstrated a significant reduction in the percentage of levofloxacin-resistant strains and a remarkable decrease in the frequency of resistance [280].

Resistance to TXA09155 did not result in cross-resistance to other antipseudomonal antibiotics, which is advantageous in clinical settings where antibiotic treatment can lead to MDR mediated by mutations in efflux pumps [282,283]. The emergence of resistance to TXA09155 was associated with specific mutations in the phoQ gene, a key component of the PhoP-PhoQ regulatory system. This system regulates the expression of over 100 genes in *P. aeruginosa*, including mexX, which is involved in efflux mechanisms and contributes to resistance against cationic antimicrobial peptides and aminoglycosides. Furthermore, TXA09155 demonstrated promising ADMET properties, indicating its potential as an effective treatment option against Gram-negative bacteria, particularly *P. aeruginosa* [136,280].

#### 4.1.3. Microbial-Derived EPIs

While most EPIs are either natural products or have originated from semi-synthetic/synthetic chemical libraries, a small proportion of them have microbial sources. Microorganisms have evolved EPIs that target MDR efflux pumps as a part of their chemical arsenal against competing species [241]. Two EPI compounds, EA-371α and EA-371δ (Figure 13), were discovered in 2001 from a library of 78,000 microbial fermentation extracts [284]. These compounds are produced by a closely related family of *Streptomyces*, including *Streptomyces velosus* [284,285]. EA-371α and EA-371δ exhibit potent and specific inhibition of the *P. aeruginosa* Mex-AB-OprM efflux pump [284]. They significantly lower the minimum potentiating concentration (MPC8) of levofloxacin by 8-fold, with EA-371α at 4.29 µM and EA-371δ at 2.15 µM, against the MexAB-OprM overexpressing strain PAM1032 [148]. It is important to note that EA-371α and EA-371δ do not possess inherent antibacterial activity against the tested *P. aeruginosa* strain [284]. EA-371α exhibited dose-dependent substrate accumulation in PAM1032, indicating its EPI activity [241,284]. However, it showed no effect in PAM1626, which lacks specific efflux pumps (*mexAB-oprM*, *mexCD-oprJ*, and *mexEF-oprN*), highlighting the importance of these pumps. Both compounds were inactive against MexCD-OprJ- and MexEF-OprN-overproducing *P. aeruginosa* strains [241,284]. Despite its cytotoxicity to mammalian cells, EA-371α’s unique structure allows for the synthesis of non-toxic derivatives with enhanced potency and bioavailability [201,284]. Computational studies using 3D crystal structures of efflux pumps can provide insights into the molecular interactions of these compounds [241,284].

Recently, a compound called ethyl 4-bromopyrrole-2-carboxylate (RP1) (Figure 14) was discovered from the soil bacterium *Streptomyces* spp. during the screening of a library containing 4000 microbial exudates [286]. RP1 demonstrated remarkable synergistic activity when combined with antibiotics, leading to a reduction in the MIC values of strains that overexpressed specific efflux pumps, namely Mex-AB-OprM in *P. aeruginosa* and AcrAB-TolC in *E. coli*. It is noteworthy that RP1 exhibited comparable EPI potency to PAβN and effectively potentiated the activity of various antibiotics, including levofloxacin, tetracycline, cloxacillin, and chloramphenicol, against both *P. aeruginosa* and *E. coli* strains, whether wild-type or overexpressing efflux pumps [286]. RP1 holds significant promise in combating bacterial MDR. Its low cytotoxicity and lack of impact on Ca^2+^ channels make it a favorable EPI candidate [286]. In vivo experiments have demonstrated RP1’s ability to reduce bacterial invasion in macrophages [286]. When used in combination with levofloxacin, RP1 exhibits remarkable efficacy and safety in treating lung infections. These findings underscore the potential of RP1 as a valuable EPI for augmenting the effectiveness of conventional antibiotics against MDR bacteria. Further investigation is warranted to fully elucidate RP1’s capabilities as an EPI [286].

All the aforementioned EPIs, introduced from a variety of sources including plants, synthetics, and microbes, each present unique properties that enable them to inhibit the function of efflux pumps. The distinct MoAs exhibited by these EPIs can be broadly classified into the following categories [123,141].

(i)Competitive inhibition: this involves EPIs that bind to the same site as the substrate and compete with it for access to the pump. This reduces the amount of substrate that can be transported out of the cell and increases its intracellular concentration [75,216]. An example of a competitive inhibitor is PAβN, which binds to the substrate-binding site of RND efflux pumps and inhibits their activity [123,221].(ii)Non-competitive substrate action: this mechanism involves EPIs that bind to a different site than the substrate on the efflux pump and prevent its transport by altering the conformation or function of the pump. These EPIs are also called substrate inhibitors because they act as substrates for the efflux pump but cannot be transported out of the cell [200,287]. An example of a non-competitive substrate inhibitor is D13-9001, which binds to a novel site on AcrB, a component of the RND efflux pump AcrAB-TolC, and inhibits its activity [221,272,273,274].(iii)Hindering functional movement: this strategy includes EPIs that bind to the efflux pump and interfere with its functional movement or rotation, which is essential for transporting substrates across the membrane [288,289]. These EPIs are also called allosteric inhibitors because they bind to a site other than the substrate-binding site and affect the activity of the efflux pump indirectly [290,291]. An example of an allosteric inhibitor is PAβN, which binds to MexB, a component of the RND efflux pump MexAB-OprM, and inhibits its activity by hindering its functional movement [221].(iv)Co-substrate action: this method incorporates EPIs that act as co-substrates for the efflux pump and require its energy-dependent transport to exert their inhibitory effect [292]. These EPIs are also called prodrugs, because they are converted into active inhibitors inside the bacterial cell after being transported by the efflux pump [293]. An example of a prodrug is MBX-3132, which is transported by RND efflux pumps and then oxidized into a reactive quinone-imine intermediate that covalently modifies and inhibits the pumps [294,295].

These diverse mechanisms are an illustration of the versatile ways in which EPIs can enhance the effectiveness of antibiotics against resistant bacteria. Each MoA carries its own potential and challenges for the design and development of potent EPIs, emphasizing the need for a comprehensive understanding of these mechanisms to effectively combat antibiotic resistance [123,218].

### 4.2. Prometheus’s Gift Unleashed: Antisense EPIs Tackle *P. aeruginosa* MDR

The utilization of the antisense approach to combat MDR bacteria signifies a paradigm shift in the field. Unlike conventional antibiotics that target proteins or macromolecular complexes, antisense oligomers offer a specific means to inhibit the expression of targeted genes, rRNA, or mRNA [296]. This innovative approach holds tremendous potential in overcoming bacterial resistance and providing effective therapeutic options. The advent of whole genome sequencing has significantly advanced our understanding of the microevolution of pathogenic bacteria and unveiled new targets for antimicrobial interventions [297]. By analyzing the complete genomes of pathogenic bacteria and studying various isolates, researchers have gained comprehensive insights into the genetic changes that underlie resistance and identified novel targets for the development of next-generation antimicrobials. The integration of the antisense approach with genomic analysis offers exciting prospects in the fight against MDR bacteria. Antisense oligomers directly interfere with specific genetic elements, disrupting critical cellular functions and rendering bacteria susceptible to treatment. Leveraging the knowledge derived from genomic analysis, tailored antimicrobial strategies can be designed to effectively combat resistance [297].

The utilization of specific antisense antibiotics presents a promising strategy in the fight against antibiotic resistance. These potential antimicrobials can be synthesized rapidly and tailored to address emerging resistance mechanisms [298]. In the context of *P. aeruginosa*’s MDR phenotype, the MexAB-OprM efflux pump, with a particular focus on the OprM component, assumes a critical role in expelling antibiotics and contributing to resistance. Manipulating the expression of OprM offers a potential avenue to disrupt efflux pump functionality and restore susceptibility to antibiotics [296]. A recent investigation explored the efficacy of an innovative delivery system employing an anion liposome [296]. This system encapsulated specific oligodeoxynucleotide (ODN) complexes, including the anti-oprM phosphorothioate oligodeoxynucleotide (PS-ODN617) and the polycation polyethylenimine (PEI). Encapsulation facilitated a notable reduction in oprM expression within MDR *P. aeruginosa* isolates, resulting in a concentration-dependent inhibition of bacterial growth when exposed to piperacillin. Additionally, this approach demonstrated the capacity to lower the MICs of commonly used antipseudomonal antibiotics against clinical *P. aeruginosa* isolates [296]. These findings support the hypothesis that targeting OprM mRNA using antisense oligomers holds substantial potential for the development of innovative therapeutics to combat emerging infectious diseases caused by *P. aeruginosa*. The antisense approach targeting OprM not only inhibits the MexAB-OprM efflux pump but also has the potential to inhibit MexXY which is involved in intrinsic resistance and efflux of various antibiotics, including aminoglycosides [296]. Thus, this approach can disrupt the function of both MexAB-OprM and MexXY efflux pumps. Further exploration of this field is warranted to overcome antibiotic resistance and effectively address the challenges posed by MDR bacteria.

### 4.3. Perseus Unleashed: Phage-Based Therapeutics against *P. aeruginosa* MDR

Lytic bacteriophages (phages) are emerging as promising agents in the search for new anti-pseudomonal drug therapies [299]. Phages possess selective bactericidal activity, targeting and killing bacteria through a lytic mechanism. Their narrow spectrum, self-amplification capability, and effectiveness against highly MDR strains make them advantageous in combating complex infections. The increasing attention and acceptance of phages highlights their potential as valuable tools in the fight against pseudomonal infections [300,301,302]. The utilization of bacteriophages (phages) as potential therapeutic agents against *P. aeruginosa* infections has been documented since 1957 [303]. Phages have been found to bind to specific targets such as lipopolysaccharide (LPS) or type IV pili, indicating that the emergence of phage-resistant mutants may result in reduced pathogenicity due to the loss of key virulence factors [303]. This vulnerability makes these mutants susceptible to existing antibiotics, highlighting the potential of phages as effective adjuvants in the treatment of *P. aeruginosa* infections [304]. 

For instance, a study by Chan et al. (2018) provided compelling evidence for the efficacy of phage therapy targeting OprM in combating *P. aeruginosa* infections. By targeting the OM component of both the MexAB and MexXY efflux systems (Figure 15), phages disrupted the tripartite Mex-efflux pump, which resulted in a significant reduction in *P. aeruginosa* populations and its frequency of resistance [305,306]. This approach not only increased the susceptibility of bacteria to antibiotics but also selected for escape mutants with heightened sensitivity to antibiotics, including ceftazidime. The findings suggest that mutations or deletions in the operon encoding the multidrug efflux pump could render bacteria more susceptible to antibiotic treatment. Such insights pave the way for the development of phage-based therapeutic strategies that consider the inevitable evolution of phage resistance, offering a promising solution to combat MDR bacterial pathogens [306].

## 5. MexB: The Zeus of Dominance—Tripartite Conquest in Bacterial Resistance

MexB, as the IM component of the MexAB-OprM tripartite system, plays a crucial role in intrinsic MDR in *P. aeruginosa*. Its substrate specificity and energy transport process contribute significantly to the efflux of a broad range of substances, including antibiotics. Studying MexB is of utmost importance in understanding and combating MDR, as it provides valuable insights into the mechanisms of drug efflux and the potential export of virulence factors, impacting colonization and infection of host cells [307].

MexB forms an asymmetrical trimer where each monomer exhibits a complex topology. It consists of a 12-transmembrane α-helix structure (TMs) spanning TM1 to TM12. Additionally, MexB possesses a substantial periplasmic domain formed by elongated loops located between TM1 and TM2, as well as between TM7 and TM8 (Figure 16). This asymmetry in the MexB trimer contributes to its functional properties and allows for efficient substrate binding and transport [307]. The periplasmic domain can be subdivided into six distinct subdomains: PN1, PN2, PC1, PC2, DN, and DC. The subdomains PN1, PN2, PC1, and PC2 work together to form the pore domain, which is responsible for creating a channel or pathway through which substances can traverse. On the other hand, the DN and DC subdomains play a key role in constructing the docking domain that interacts with the OMF. This docking domain facilitates the binding and interaction of MexB with the OMF, connecting IM to OM. By having these specific subdomains within its periplasmic domain, MexB is able to carry out its function in substrate transport and efflux with enhanced efficiency and effectiveness [307]. The DN subdomain of MexB exhibits an elongated β-sheet that extends into the neighboring subunit, promoting interlocking and stability within the trimeric structure. This inter-subunit interaction is vital for maintaining the structural integrity and resilience of MexB [308]. Moreover, a pseudo-two-fold symmetry can be observed in the transmembrane region, where TM1–TM7 correlates with TM8–TM12. Notably, TM4 and TM10, surrounded by other transmembrane segments, contain highly conserved charged residues, including Asp407, Asp408 (TM4), and Lys939 (940 in AcrB) (TM10). These charged residues likely play a significant role in facilitating proton translocation, a critical process for MexB’s proper functioning. Mutations affecting these residues have been found to result in complete loss of drug resistance [308].

The comparative analysis of MexB and AcrB provides valuable insights into their structural similarities and distinctions. MexB, composed of 1046 amino acids, exhibits two distinctive internal amino acid deletions at positions 711 and 955. These deletions occur in the PC2 loop and the cytoplasmic loop, connecting TM10 and TM11, respectively. In contrast, AcrB, with 1049 amino acids, lacks these deletions. These subtle variations in their molecular composition contribute to the unique functional characteristics and transport capabilities of each pump [307] (Figure 17). Understanding these nuances enriches our scientific exploration of these remarkable transporters.

With a sequence identity of 69.8% and amino acid similarity of 83.2% in a 1046-residue sequence, MexB closely resembles its *E. coli* counterpart AcrB [309,310]. Additionally, they exhibit similar substrate specificities. However, the MexAB–OprM system demonstrates a distinct resistance pattern compared to the AcrAB–TolC system when expressed in *E. coli* [311]. The MexAB–OprM complex, which is a type of efflux pump, shows higher resistance to cinoxacin but provides less protection against EtBr, oleandomycin, and erythromycin compared to another efflux pump called AcrAB–TolC [250]. Despite MexB and AcrB sharing a high degree of similarity in their protein sequences, they cannot be interchanged without losing their activity because they interact differently with other components in the system [311,312]. Interestingly, while the OMFs TolC and OprM can work with multiple efflux pumps, MexB and AcrB have unique roles in detecting, interacting with, and expelling specific substances [55,313].

The structural similarities between MexB and AcrB suggest that EPIs against MexB may also inhibit the activity of AcrB in *E. coli* [137]. This has significant implications for combating antibiotic resistance, as it opens up the possibility of developing EPIs that can target multiple efflux pump systems found in different Gram-negative bacteria. AcrAB-TolC homologs, which are similar efflux pump systems, are distributed widely among Gram-negative bacteria [314]. Leveraging the structural similarity between MexB and AcrB, researchers can design broad-spectrum EPIs that have the potential to restore the activity of a wider range of antibiotics, making them more effective against various bacterial infections [241].

The crystal structures of MexB have provided important insights into the process of drug extrusion, revealing three distinct states: loose (L), tight (T), and open (O) [315] (Figure 18). Despite having identical sequences, each of the three monomers exhibits unique conformations. Within MexB, there are two significant binding sites. The proximal pocket is located in the L monomer, while the T monomer contains a deep binding pocket (DP_T_), also known as the distal cavity [316]. The DP_T_ serves as a crucial passage for compounds being transported by MexB, featuring a cluster of hydrophobic residues, including phenylalanine, creating a hydrophobic trap (HT) [317,318]. The DP_T_ can be further divided into three distinct subregions: an interface region that separates it from the proximal pocket and includes the switch loop [319], a spacious cave region, and a narrow groove in its deeper portion (Figure 18) [315]. This detailed structural knowledge sheds light on the intricate mechanisms of drug binding and extrusion facilitated by MexB.

The transmembrane domain of AcrB/MexB encompasses a central hole surrounded by three bundles of 12 α-helices, forming a structurally fascinating arrangement (Figure 19). Contrary to expectations, this central hole is not filled with water but with a phospholipid bilayer, creating a unique environment within the protein [61,318].

Within this phospholipid bilayer, there exists a central cavity that plays a critical role in substrate binding and transport (Figure 20A). The cavity features three windows, known as vestibules, located on the surface of the inner membrane (Figure 19 and Figure 20B). Originally, this central cavity was identified as the site where substrates enter the protein through the vestibules, binding to the cavity, traversing the open central pore, and finally being expelled through the funnel-shaped exit at the top of the AcrB/MexB trimer into the TolC/OprM channel [61,318].

The unique architecture of the transmembrane domain, with its central hole and phospholipid-filled cavity, provides an intriguing framework for substrate binding and transport. The vestibules on the membrane surface serve as entry points for substrates to access the central cavity, which acts as a hub for substrate interaction. When the central pore is open, substrates can pass through it, ultimately being expelled through the unique funnel-like exit situated at the apex of the AcrB/MexB trimer. This elaborate process highlights the intricate interplay between the protein and the phospholipid bilayer, demonstrating the sophisticated machinery employed by AcrB/MexB to facilitate the transport of substrates across the cell membrane [61,307,318].

## 6. Unveiling the Mythological Enigma: MexB’s Multifaceted Binding Sites for Diverse Ligands

### 6.1. Multiple Binding Pockets and Ligand Interactions of MexB

Within the MexB pump, researchers have identified unique drug-binding pockets referred to as the proximal (PBP) and distal (DBP) binding pockets. These distinct pockets serve as sites for the interaction and binding of various compounds. The PBP and DBP play crucial roles in substrate recognition and transport within the MexB pump [316]. The DBP, characterized by an EPI-binding hydrophobic pit, serves as a site for specific ligands, including pyridopyrimidine and pyranopyridine derivatives. These EPIs exhibit remarkable affinity for the pit, contributing to their potent inhibitory properties [318,320]. Notably, recent studies have revealed the presence of an additional binding site located near the periplasmic surface within the transmembrane region. This novel binding site expands the repertoire of substrate recognition and binding within MexB, offering new insights into the intricate mechanisms of substrate transport and resistance [321].

Further investigation into the distal pocket of MexB has revealed its ability to accommodate larger compounds, such as lauryl maltose neopentyl glycol (LMNG) (Figure 21), a surfactant that competes with other substrates for export [322]. The crystal structures of MexB have demonstrated the binding of LMNG and other macromolecular compounds such as neopentyl glycol derivative C7NG within the distal pocket, elucidating the molecular interactions governing their recognition and transport. The fascinating interplay between molecular weight and individual molecular characteristics highlights the complexity of substrate binding in the distal and proximal pockets [322].

The inhibitory activity of the substrates is deeply related to the insertion of alkyl chains, which are hydrophobic chemical groups composed of carbon and hydrogen atoms, into the hydrophobic pit located in the distal binding pocket [322]. These alkyl chains, present in specific substrates, play a critical role in interacting with the hydrophobic environment of the pit. The insertion of alkyl chains into the hydrophobic pit allows for favorable molecular interactions, enhancing the inhibitory properties of these substrates. The extent and nature of these interactions influence the binding affinity and effectiveness of the substrates in inhibiting the transporter’s function [322]. Therefore, the specific characteristics of the alkyl chains and their compatibility with the hydrophobic pit are essential factors governing the inhibitory activity of these substrates.

### 6.2. Fluoroquinolone Antibiotic Binding Sites in MexB

Molecular recognition among protein and ligand complexes is driven by intermolecular forces, including interactions such as van der Waals forces and hydrophobic interactions. Shape complementarity, an essential aspect of molecular recognition, refers to the geometric matching between the surfaces of the protein and its ligand. It ensures a precise fit between their three-dimensional shapes, maximizing surface contact and facilitating stable and specific protein–ligand complexes. Shape complementarity relies on various non-covalent bonds, including polar interactions such as electrostatic interactions and hydrogen bonds, as well as apolar interactions such as van der Waals forces and hydrophobic interactions [315]. In order to comprehensively understand the intricate mechanisms underlying *P. aeruginosa* MDR and devise effective strategies to combat it, it is imperative to explore the molecular intricacies of target protein binding pockets and ligand interactions [323]. Within this context, FQs emerge as a particularly significant class of substrates, given their status as the preferred substrates for almost all Mex efflux pumps [148]. This association is noteworthy due to its implications in the emergence of highly MDR and even PDR bacterial isolates, primarily driven by the overexpression of these efflux systems in response to FQ exposure [148]. This, in turn, leads to a worrisome cross-resistance phenomenon, encompassing a wide range of substrates transported by Mex efflux pumps [148]. Moreover, it is worth highlighting that FQs hold a prominent position in the treatment of Gram-negative infections, further emphasizing the significance of studying their interactions with Mex efflux pumps [324]. By gaining a deeper understanding of the binding modes and mechanisms underlying FQ interactions with Mex efflux pumps, we can pave the way for the development of innovative approaches to tackle drug resistance effectively.

The availability of crystal structures provides invaluable insights into the molecular interactions between efflux pumps and antibiotics, aiding in our understanding of drug resistance mechanisms. For instance, crystal structures of the AcrB efflux pump bound to antibiotics from various classes have been extensively studied [325]. In addition to previously reported crystal structures, a recent study has revealed the binding of levofloxacin to the DP_T_ of AcrB, providing further insights into efflux mechanisms of this pump [264]. On the other hand, in the case of MexB, only two crystal structures have been reported so far, featuring the binding of the inhibitor D13-9001 and the substrate LMNG [315,318]. To gain a more comprehensive understanding of MexB and its binding capabilities, it becomes crucial to employ computational studies and similar investigations that can uncover detailed structural information pertaining to the binding of diverse compounds to MexB [326,327].

Gervasoni et al. (2022) conducted an extensive investigation into the docking poses and overall complementarity between 36 distinct FQs (Figure 22) and the multidrug efflux transporter MexB. The study focused on the intricate binding pocket of MexB and explored how the FQ compounds interacted with it. The researchers discovered that the FQs exhibited diverse binding mechanisms within MexB’s pocket, highlighting the broad flexibility of MexB in accommodating different compounds [328]. This finding supported the diffusive binding (or oscillation) hypothesis, which suggests that MexB substrates can dynamically transition between various binding modes [61,329]. Interestingly, despite the diverse binding mechanisms, the FQ compounds maintained comparable affinities within the DP_T_. To provide convenient access for further analysis, the study made the predicted binding modes of each FQ compound available as downloadable PDB files. Researchers can access and visualize these files online using the provided web address: https://www.dsf.unica.it/dock/mexb/quinolones (accessed on 6 August 2023) [315]. This resource enables researchers to explore the specific binding interactions between MexB and each FQ compound, enhancing our understanding of the molecular recognition process.

Gervasoni et al. also made notable observations regarding the binding preferences of different compounds within the binding pocket of MexB. Specifically, they found that levofloxacin and D13-9001 exhibit a higher propensity to bind within a specific region referred to as the “cave”, while minocycline shows a preference for binding to a different area known as the “groove” (Figure 23). These findings suggest that the binding affinity of these compounds is influenced by shape and surface complementarities, as indicated by the surface-matching data and the overall trend of the docking scores [315].

Their study uncovers the intriguing binding preferences of FQs towards MexB, a pivotal component in the development of multidrug resistance [315]. Notably, the majority of FQs demonstrate a pronounced affinity for the groove region within MexB’s binding pocket, while flumequine and sarafloxacin exhibit a distinct inclination towards the cave region [315]. These findings challenge the notion of a uniform binding pattern and underscore the influence of subtle chemical variations on the diverse binding modes observed [221,293]. Significantly, the groove region emerges as the primary hub of FQ interaction, displaying higher occupancy compared to the interface and cave regions, which exhibit comparable preferences [315]. These observations suggest the potential existence of multiple binding modes influenced by protonation states within the distal pocket, highlighting the intricate relationship between compound structure and binding behavior [61]. With the focus on levofloxacin as the reference compound, alongside its co-structure with AcrB, MexB’s homolog, the study paves the way for further investigations into the dynamic interplay between FQs and MexB, contributing to a deeper comprehension of mechanisms underlying multidrug resistance [315].

### 6.3. EPI Binding Sites in MexB

In a seminal study by Nakashima et al. (2013) [318], the researchers investigated the structural basis of inhibitor specificity in MexB using the crystal structure analysis of MexB bound to D13-9001 (Figure 24). The crystal structures provided a detailed atomic-level understanding of the interaction between MexB and D13-9001, revealing the specific molecular interactions at the binding site [318]. Interestingly, the binding site in MexB showed some similarity to the binding site found in AcrB, suggesting a conserved mechanism across different efflux pumps [330]. D13-9001 consisted of a hydrophobic TAP moiety composed of four aromatic rings attached to a phosphonium ion [331]. Within the crystal structure, the hydrophobic TAP moiety was observed to be inserted into a narrow hydrophobic trap formed by specific amino acid residues [332]. These residues included phenylalanine (F) at positions 136, 178, 610, 615, 628, and 573 [318,333]. Another component of D13-9001 was the hydrophilic tetrazole ring, which interacted with specific amino acids within the binding site. These amino acids included aspartic acid (D) at position 274, arginine (R) at position 620, and lysine (K) at position 151. These interactions played a crucial role in stabilizing the binding of D13-9001 to MexB and conferring its specificity [318,334].

One notable difference between MexB and AcrB was observed in the conformation of the PAEA moiety, a component of D13-9001. In MexB, the PAEA moiety extended towards arginine (R) 128, while in AcrB, it adopted a distinct bent structure between the piperidine ring and the aceto-amino ethylene ammonio-acetate group. This conformational difference highlighted the structural diversity among related proteins [318]. The structural organization of MexB’s distal pocket, illustrated in Figure 24d–f, shows that it is divided into two distinct regions. The substrate-translocation channel, relatively hydrophilic in nature, provided ample space for simultaneous accommodation of multiple drug molecules [318,335]. In contrast, the hydrophobic trap formed a deep and narrow fissure, facilitating the binding and sequestration of hydrophobic compounds [336]. Thus, structural analysis of the inhibitor-bound crystal structures of AcrB and MexB reveals that the pyridopyrimidine derivative D13-9001 binds to a narrow hydrophobic pit in the distal pocket, interacting with Phe178 and disrupting the functional rotation cycle, thus providing the basis for its inhibitory activity [318,337].

Another interesting finding is that the pyridopyrimidine derivatives selectively inhibit AcrB and MexB while sparing MexY. The crystal structures reveal a narrow hydrophobic trap in the distal pocket, formed by a phenylalanine cluster, which serves as the binding site for the inhibitor [318]. This impedes the functional rotation mechanism. The interaction between phenylalanine (Phe) 178 and the inhibitor is crucial for its strong binding [336]. In contrast, the bulky side chain of tryptophan (Trp) 177 in MexY prevents inhibitor binding [307]. These findings have implications for the development of universal inhibitors targeting MexB and MexY, offering potential solutions to combat multidrug resistance in Gram-negative pathogens [22,318].

In a complementary study led by Yamaguchi et al. (2015), it was found that the hydrophilic component of D13-9001 binds to the identical site as minocycline and doxorubicin antibiotics [61]. By introducing the F610A mutation in this site, substrate penetration is enabled, resulting in a decline in export activity. Furthermore, the presence of Phe178 at the site periphery facilitates π–π interactions with D13-9001, thereby establishing stable binding and exerting inhibitory effects (Figure 25A,B). These interactions disrupt the functional rotation cycle necessary for the transition to the extrusion stage [61].

In the MexY homology model, the presence of a bulky indolyl side chain of tryptophan (Trp177) at the corresponding pit position hinders the binding of D13-9001 (Figure 25C). However, introducing a phenylalanine substitution (W177F) in MexY (Figure 25D) resulted in a mutant that exhibited susceptibility to D13-9001 comparable to AcrB, while maintaining drug export activity [61]. Conversely, substitution of tryptophan for Phe178 in AcrB (AcrB F178W) conferred resistance to D13-9001 similar to MexY. The crystal structure analysis of the AcrB F178W mutant unveiled the distinctive conformation of the indolyl side chain of Trp178 extending into the pit (Figure 25E), offering crucial insights into the structural mechanisms underlying resistance to D13-9001, akin to MexY [61].

This indicates that the specificity of D13-9001 is determined by the presence of a bulky side chain at either position 178 or 177 [61]. Notably, the MexB F178W mutant exhibits sustained sensitivity to D13-9001, challenging previous expectations. In the crystal structure of the D13-9001-bound MexB F178W mutant, an intriguing binding configuration is observed, where the protruding indolyl side chain aligns parallel to the pit’s inner wall. This arrangement promotes stable binding through π–π interactions with the pyridopyrimidine ring (Figure 25F) [61,338,339], providing novel insights into the binding mechanism of D13-9001.

Through extensive computational simulations, it was revealed that the parallel positioning of the indolyl moiety of Trp178 in AcrB is hindered by steric interactions with Val139 [61,339,340]. In contrast, MexB, which possesses a slightly larger binding pocket, allows for this specific orientation. Similarly, the presence of Ile138 in MexY also obstructs the parallel alignment of Trp177 [61,293,318]. To confirm these findings, novel double mutants, namely AcrB F178W V139A and MexY I138A, were generated (Figure 25G,H). Interestingly, both mutants demonstrated susceptibility to D13-9001, resembling the drug sensitivity observed in the wild-type AcrB protein [61,329].

The distinct characteristics exhibited by the hydrophobic pit situated in the distal binding pocket play a pivotal role in determining the specific interactions of pyridopyrimidine derivatives, including D13-9001 [61,221,322]. Notably, the binding structures of D13-9001 with AcrB and MexB have revolutionized our understanding by providing unprecedented insights into the inhibitory mechanisms of multidrug efflux transporters in physiologically active asymmetric forms [274,320,341]. These remarkable structures have laid a solid foundation for the development of universal inhibitors that can selectively target AcrB, MexB, and MexY, thereby opening new avenues for effective therapeutic interventions [61,318].

## 7. Discussion

In this review, we have provided a comprehensive overview of the clinical significance of efflux pumps, particularly RND efflux pumps, in *P. aeruginosa* strains. We have discussed the strategies employed to counteract these pumps and highlighted the diverse types of EPIs available. A major focus has been given to the MexAB-OprM pump, one of the most prominent RND pumps. We have examined the substrate binding sites of MexAB-OprM and explored the inhibitory mechanisms through the analysis of co-crystallographic structures and molecular dynamics simulations. Moreover, we have discussed the increasing structural diversity of EPIs and their ability to target specific binding pockets, which holds great promise for the discovery of potent EPIs. These advancements pave the way for future clinical applications [315,316,318,323,329,342].

The development of efficient EPIs is a challenging task due to the complex relationship between antibacterial compounds and efflux pumps [51]. MDR efflux pumps, particularly the RND pumps, possess large binding sites capable of accommodating multiple compounds owing to their broad substrate specificities [61]. However, identifying potent EPIs is further complicated by the absence of specific molecular properties such as molecular weight, logP (partition coefficient), net charge, or polar surface area (PSA) that can definitively classify a compound as an efflux pump substrate or an EPI [241]. Interestingly, a compound initially identified as an efflux pump substrate, preferentially expelled from bacteria, may unexpectedly exhibit potent EPI activity. This intriguing mechanism, known as “competitive substrate inhibitors”, not only inhibits efflux pump activity but also holds the potential to restore the efficacy of antibiotics when co-administered [241,343]. To develop highly potent EPIs, it is crucial to initiate the screening of ligands (potential EPIs) with small molecules that exhibit physicochemical properties in accordance with Lipinski’s rule of five, which evaluates the drug-likeness of compounds [344]. This rule suggests that successful EPIs should possess specific characteristics, including a molecular weight of 300 or less, a clogP (lipophilicity) value of 3 or lower, a PSA of 3 or less, a maximum of three rotatable bonds, and no more than three hydrogen bond donors and acceptors [229]. Focusing on these criteria, particularly when considering small molecules during the screening process, can aid in identifying potential EPIs that possess the necessary properties for efficient inhibition of efflux pumps [280].

Additionally, it is crucial to extend optimization efforts beyond enhancing potency against the target efflux pump and prioritize the assessment of the ADMET profile of the EPI before proceeding to clinical trials. This would help ensure the safety and efficacy of the EPI in a clinical setting [123,345]. Therefore, it is essential to conduct SAR studies to identify and address any challenges related to permeability and solubility [346] which would enhance the properties of the EPI, ultimately improving its efficacy.

The comparison of potent Gram-negative EPIs, such as PAβN (Figure 4), D13-9001 (Figure 10), and MBX2319 (Figure 11), highlights remarkable structural similarities in their chemical scaffolds. These EPIs demonstrate a consistent presence of at least two hydrophobic ring systems that interact with hydrophobic amino acid residues within the substrate binding site [275]. Potent EPIs demonstrate remarkable structural features that contribute to their effectiveness in addressing antibiotic resistance. These shared characteristics include the presence of diverse ring moieties, such as quinoline, quinolone, benzene, pyridine, pyranopyridine, pyrimidine, pyridopyrimidinone, or indole. These rings facilitate interactions with hydrophobic amino acid residues in the substrate binding site, enhancing the efficacy of EPIs. Moreover, electron withdrawing groups, such as halogens (e.g., fluorine, chlorine, bromine, or iodine), cyano group (CN), and nitro group (NO_2_), are often incorporated into the ring systems. These electron withdrawing groups are associated with the inhibitory activity of EPIs against efflux pumps [241]. Additionally, EPIs exhibit the presence of at least two hydrophobic ring systems, with at least one ring containing a nitrogen atom. This structural arrangement contributes to the improved binding affinity of EPIs. Furthermore, methoxy groups are frequently observed, primarily attached to the benzene or quinoline ring, which play a role in shaping the overall molecular structure. Lastly, EPIs possess side chains that contain basic amine groups, including secondary, tertiary, or primary amines. These side chains influence the potency and specificity of EPIs as inhibitors. The combination of these unique structural elements underscores the distinctiveness and potential of EPIs in combatting antibiotic resistance [241].

A salt-bridged network involving Asp407, Asp408 in TM4, and Lys939 in TM10 has been identified in *P. aeruginosa*. This network is stabilized by hydrogen-bonding interactions with Thr976 in TM11 [332]. Indeed, TM4 has been recognized as a major pathway for proton translocation. It plays a crucial role in the transport mechanism of efflux pumps such as AcrB and MexB [332]. A plausible hypothesis suggests that upon exposure of efflux pumps to drugs, there is a notable interaction between a proton and the Asp407 residue. This interaction perturbs the stability of the salt-bridge and hydrogen-bonding network, leading to the influx of protons into the cytoplasm. Consequently, this proton influx triggers the efflux of drugs from the cell [332].

AcrB, a close homologue of MexB, possesses three potential drug-binding sites crucial for its efflux pump activity. These sites include an extensive central cavity known for its remarkable size and capacity to accommodate various drug molecules [309]. Additionally, there is a deep external depression situated on the outer surface of the C-terminal periplasmic loop domain, providing an additional binding pocket that facilitates interactions with specific drugs [347]. Furthermore, a deep inside periplasmic domain contributes to the efflux process by offering a distinct binding site within the protein structure. These three distinct drug-binding sites collectively enhance the recognition, binding, and removal of diverse drug compounds, playing a vital role in the resistance mechanism mediated by efflux pumps in bacteria [61,318,332]. In the initial stage of substrate recognition, the central cavity of RND pumps, formed by the TM domains of the three protomers, plays a crucial role. This central cavity is located at the interface between the periplasm and the inner membrane [309]. A co-crystallography study conducted on AcrB with four structurally diverse ligands revealed that during substrate binding, these ligands predominantly interact with the periplasmic loop regions located between the TM3 and TM4 loops, as well as between the TM5 and TM6 loops within the upper region of the central cavity [309]. This indicates that the initial binding of substrates occurs in specific regions of the central cavity, highlighting the importance of these periplasmic loop regions in the substrate recognition process. The TM5 and TM6 regions of RND pumps are known to contain four highly conserved phenylalanine residues, namely, Phe386, Phe388, Phe458, and Phe459, which play a critical role in the primary interactions between substrates and the pumps [309]. In a study involving an AcrB N109A mutant bound to similar substrates, it was observed that most ligands were closely associated with the top regions of TM5 and TM6. However, PAβN was found to be attached slightly to the left of the other ligands, towards the TM3 and TM4 regions [347]. This suggests that the binding orientations and preferences of different substrates within the TM5 and TM6 regions may vary, with PAβN showing a distinctive binding location compared to other ligands.

The binding sites in MexB were similar to that of AcrB, suggesting a conserved mechanism among efflux pumps. The EPI D13-9001 consists of a hydrophobic TAP moiety and a hydrophilic tetrazole ring. The hydrophobic TAP moiety interacts with specific amino acid residues, including phenylalanine (F) at positions 136, 178, 610, 615, 628, and 573. The hydrophilic tetrazole ring interacts with aspartic acid (D) at position 274, arginine (R) at position 620, and lysine (K) at position 151, stabilizing the binding of D13-9001 to MexB [318]. The crystal structures also revealed a distinct conformational difference in the PAEA moiety of D13-9001 between MexB and AcrB. MexB’s binding pocket is divided into two regions: a relatively hydrophilic substrate-translocation channel and a deep and narrow hydrophobic trap. The hydrophobic trap, formed by a phenylalanine cluster, facilitates the binding of hydrophobic compounds. These structural insights provide a basis for understanding EPI binding to MexB and offer potential strategies for the development of universal inhibitors targeting multidrug efflux transporters [61,318,332].

The involvement of TM4 in substrate binding plays a critical role in the functioning of RND pumps. Normally, the α-helix structure of TM4 maintains regular hydrogen-bonding interactions between the backbone amine functional (NH) and carbonyl functional (C=O) groups. However, when TM4 is unusually engaged in substrate binding, these hydrogen-bonding interactions are disrupted. This disruption leads to the separation of Asp407 and Asp408 from the rest of the salt bridge/H-bond network, which includes Lys939 and Thr978. As a result, the network loses its ability to be protonated, which is essential for supplying energy to RND pumps. Since drug efflux is coupled with proton influx, the disturbance of proton translocation hinders drug efflux to the external medium. Without drug efflux, the function of the pump is inhibited [334,348]. Additionally, the binding of drugs to RND pumps initiates a series of conformational changes that ultimately enable drug efflux. These changes are initiated by the acquisition of protons through a proton relay network. Here, PAβN acts as an EPI, and its mechanism of action involves disrupting the regular α-helix structure in TM4. This disruption results in the interference of the proton relay network, preventing the subsequent conformational changes required for proper substrate binding. One intriguing aspect is the interaction between the carboxylate oxygen of Phe396 and the amino group of phenylalanine in PAβN. This interaction suggests that the amino acid located in the first section of PAβN’s structure plays a vital role in the activity of the EPI [148,348]. Overall, the involvement of TM4 in substrate binding and the disruption of the proton relay network by EPIs such as PAβN have significant implications for the functioning and inhibition of RND pumps. Further research is necessary to fully elucidate the precise mechanisms and optimize the design of effective EPIs.

Developing effective EPIs in Gram-negative bacteria presents a complex challenge due to the divergent preferences of membrane porins and RND efflux pumps towards distinct molecule types. Membrane porins, acting as gatekeepers, predominantly facilitate the entry of smaller hydrophilic molecules through the OM. On the other hand, RND efflux pumps specialize in recognizing and expelling larger hydrophobic molecules, contributing to MDR phenotype [241].

The selectivity of membrane porins for smaller hydrophilic molecules is determined by both their structural characteristics and the functional requirements of the cell [349,350]. This selectivity can be dissected into three primary factors: (i) Structural characteristics: porins typically consist of a β-barrel structure that forms a water-filled channel, allowing the diffusion of hydrophilic substances. The size of the channel restricts the passage to only small molecules. The pore’s internal surface is often lined with polar amino acids that preferentially interact with hydrophilic molecules, further facilitating their diffusion through the channel [351,352]. (ii) Functional requirements: the selectivity of porins for smaller, hydrophilic molecules allows essential nutrients (e.g., sugars, ions, and amino acids), which are generally hydrophilic and small in size, to enter the cell. At the same time, this selectivity helps prevent the entry of larger, potentially harmful substances, such as toxins or larger hydrophobic drugs, thereby playing a crucial role in the cell’s survival and defense mechanism [118,353]. (iii) Energy considerations: passive diffusion through porins is an energy-efficient method of transport. Hydrophilic molecules can easily traverse these water-filled channels due to their compatibility with the aqueous environment inside the porin which collectively influence the development of effective EPIs [75,354].

This inherent contrast in molecular selectivity poses a rational dilemma when designing EPIs that can simultaneously penetrate the OM and inhibit efflux pumps. The preference of membrane porins for smaller hydrophilic molecules is essential for maintaining cell viability and allowing the influx of necessary nutrients. However, it also poses a challenge for EPIs targeting RND efflux pumps, as these inhibitors typically exhibit larger sizes and hydrophobic properties to interact with the pump’s binding sites effectively. Overcoming this molecular dichotomy requires innovative strategies that can bypass or circumvent the OM barrier while still effectively inhibiting the efflux pumps. Researchers are actively exploring various approaches to address this challenge. One promising avenue involves the development of EPIs that possess both hydrophilic and hydrophobic characteristics, striking a balance between penetrating the OM and interacting with the efflux pumps. Additionally, the identification of specific mechanisms involved in OM permeability and the interplay between porins, and efflux pumps is helping to guide the rational design of EPIs that can effectively disrupt drug efflux.

One of the solutions to overcome the OM permeability barrier is designing dual hydrophilic/hydrophobic EPIs, leveraging transporter proteins, or other membrane-disrupting strategies, to bypass the porin barrier. In-depth studies on the structure–function relationship of porins and efflux pumps, their interplay, and how they contribute to the bacterial resistance phenotype are also crucial to guide the rational design of more effective EPIs. Designing EPIs with both hydrophilic and hydrophobic characteristics presents a series of nuanced challenges. Primarily, striking the correct balance between these properties is crucial for the molecule to penetrate the lipid bilayer of the outer membrane and engage with the efflux pumps within the aqueous environment of the bacterial cell [123]. The inclusion of hydrophilic and hydrophobic groups necessitates distinct chemical reactions, thereby complicating the synthetic process and the molecule’s overall design and purification [147,293]. Furthermore, the dual-natured molecule may face stability issues due to partitioning into diverse environments, altering its conformation and potentially impairing its ability to bind and inhibit the efflux pump [355]. The pharmacokinetic properties, including ADMET can be significantly influenced by the balance between hydrophilicity and hydrophobicity, impacting the molecule’s bioavailability and efficacy [356]. In addition to these challenges, the EPIs need to be carefully designed to minimize interactions with other cellular components to avoid off-target effects [357]. Despite these hurdles, the development of EPIs with dual hydrophilic and hydrophobic properties holds substantial promise, contingent upon careful design, meticulous synthesis, and comprehensive characterization [22].

In theory, an effective EPI should inhibit efflux pumps without compromising the integrity of the OM. Damaging the OM could lead to the release of potentially harmful substances such as lipopolysaccharide (LPS) endotoxins, thus escalating the severity of bacterial infections [22,351]. There are some novel strategies for EPIs for bypassing or circumventing the OM barrier while effectively inhibiting efflux pumps. They are as follows: (i) Conjugation to nutrient molecules: Iron is a vital nutrient for bacteria, so they have evolved mechanisms to take up iron from the environment. Li et al. (2015) exploited this by conjugating an EPI to iron chelators, which are molecules that bind to iron. The bacteria transport the chelators across the OM, along with the attached EPI. This allows the EPI to reach the periplasmic space, where it can inhibit efflux pumps without damaging the OM [22]. (ii) Prodrug strategy: the prodrug strategy involves creating chemically altered versions of a drug that only activate once inside the body. This approach can enhance the ability of EPIs to cross the bacterial OM by creating a lipophilic prodrug that can diffuse through the OM, reach the periplasmic space, and then be converted back to its active form by bacterial enzymes, thus increasing the intracellular concentration of the EPI [123,358]. However, specific implementation of this strategy with EPIs is currently unreported, potentially due to challenges in designing a prodrug that can be selectively activated by bacterial enzymes and not mammalian ones, as well as the lack of specific and potent EPIs to target various types of efflux pumps [147,223,359]. (iii) Nanoparticle carriers: nanoparticles are tiny particles that can be engineered to carry drugs and deliver them to specific targets. They could be designed to interact with receptors on the bacterial surface and trigger transport mechanisms, delivering the EPIs into the bacterium without compromising the outer membrane [360]. For example, silver nanoparticles (AgNPs) have been shown to inhibit the MexAB-OprM efflux pump in *P. aeruginosa* and enhance the antibacterial activity of ciprofloxacin [361]. Another example is gold nanoparticles (AuNPs), which have been conjugated with PAβN and demonstrated to increase the efficacy of tetracycline against *S. aureus* [362,363]. (iv) Targeting porins: porins mainly allow small, hydrophilic molecules to pass through the OM of Gram-negative bacteria. EPIs could be designed to mimic these molecules, enabling them to enter bacteria via these channels and inhibit the efflux of antibiotics. While no EPIs have been specifically designed this way to date, some natural compounds have been shown to use this entry route and enhance the activity of antibiotics [123,364]. Moreover, some antibiotics, such as tetracyclines and β-lactams, are known to penetrate the OM through porins [365,366]. (v) Small molecule adjuvants, nanobodies, or aptamers: small molecules, nanobodies (small antibodies), or aptamers (short nucleic acid sequences) could be used to either inhibit efflux pumps or increase the permeability of the OM [367,368]. For instance, the small molecule MBX2319 inhibits a major efflux pump in *P. aeruginosa* and *E. coli* [221,277,278,279] and nanobodies against the MexAB-OprM have been developed and shown to enhance the activity of antibiotics against *P. aeruginosa* [369]. However, these approaches must be carefully controlled to avoid toxic or off-target effects.

Researchers have sought alternative mechanisms to overcome the limitations posed by membrane porins in the development of EPIs for Gram-negative bacteria. One such mechanism is the utilization of simple diffusion through the lipid bilayer of the OM. By incorporating charged groups into the structure of EPIs, they can interact with the lipid bilayer, facilitating their passage through the OM via electrostatic interactions. These charged groups enable EPIs to overcome the barrier presented by the hydrophobic core of the lipid bilayer [44,370]. The inclusion of charged groups in EPIs offers the potential to bypass the stringent selectivity of porins and gain access to the periplasmic space, where the RND efflux pumps are located. This rational design strategy aims to enhance the structure of EPIs and augment their ability to inhibit efflux pumps by improving their membrane penetration capabilities. It is crucial to note that while the incorporation of charged groups enhances membrane penetration, the overall design of EPIs is a multifaceted endeavor that necessitates meticulous consideration of factors such as their interaction with efflux pump binding sites and their overall effectiveness in combating multidrug resistance [371]. Therefore, the rationale behind incorporating charged groups into the structure of EPIs to facilitate interaction with the lipid bilayer and overcome the limitations of porins highlights the need for innovative strategies to bridge the molecular dichotomy between membrane porins and efflux pumps.

To advance our understanding of EPIs and their interactions with efflux pumps, further exploration using computational studies and complementary biophysical techniques is crucial. Traditional in vitro and in vivo assays have limitations, and therefore, alternative approaches are necessary. Computational studies, such as in silico modelling, provide valuable insights into the structure, dynamics, and molecular interactions of EPIs with efflux pumps. By simulating these interactions, researchers can determine precise ligand–protein connections, shedding light on the binding modes and affinity of EPIs. Complementary biophysical techniques such as native mass spectrometry (MS) and nuclear magnetic resonance (NMR) offer experimental validation and provide additional information on the complex formation and dynamics of EPIs and efflux pumps. Crystal structures and homology models play a significant role in deepening our understanding of the structure and function of MDR efflux pumps. By examining these structures, researchers gain insights into the binding sites and conformational changes involved in efflux pump inhibition. This knowledge not only offers valuable insights into the design and MoA of EPIs but also lays the foundation for molecular interactions of EPI-efflux pumps in order to develop clinically effective EPIs.

The understanding of efflux pump inhibition, including the determination of binding sites and the conformational changes involved, is greatly enhanced by the combined use of crystal structures, homology models, native MS, and NMR. Crystallography helps visualize the overall architecture of efflux pumps, revealing potential drug binding sites and conformational changes involved in drug efflux [75]. When crystal structures are unavailable, homology modeling serves as a useful tool, leveraging known structures of similar proteins to predict the structure and potential binding sites of the efflux pumps [372]. Native MS provides a powerful approach to studying efflux pump inhibition by preserving non-covalent interactions of proteins and their ligands under near-native conditions. This allows the identification of binding sites and accurate determination of the stoichiometry and binding affinities of protein–ligand complexes, providing valuable insights into the mechanisms of efflux pump inhibition [373,374]. NMR provides detailed information on protein structures and dynamics in solution, enabling the mapping of ligand binding sites and insights into associated dynamics and conformational changes [375,376,377]. By integrating these methods, a comprehensive understanding of efflux pump structure, dynamics, and inhibitory interactions can be obtained, aiding the development of novel EPIs.

Structure-based approaches hold immense potential in the discovery and characterization of novel EPIs by providing detailed insights into the spatial arrangement of atoms within efflux pumps. These insights enable researchers to design and optimize EPIs with precise targeting, high binding affinity, and better efficacy, thereby paving the way for effective therapeutic interventions against drug resistance [123,378]. Furthermore, the use of such approaches not only helps in understanding the interaction between EPIs and their target efflux pumps but also uncovers conformational changes upon binding, helping to elucidate the mechanisms of efflux pump inhibition, which is crucial for the development of more potent and specific inhibitors [201,221].

In summary, further computational studies and complementary biophysical techniques are essential to uncover the intricate details of EPIs’ interactions with efflux pumps. Crystal structures, homology models, and a structure-based approach offer valuable insights into the design of clinically efficacious EPIs. By pursuing these avenues, researchers can identify novel EPIs, elucidate their MoA, and gain a deeper understanding of the complex ligand–target interactions involved.

## Figures and Tables

**Figure 1 antibiotics-12-01304-f001:**
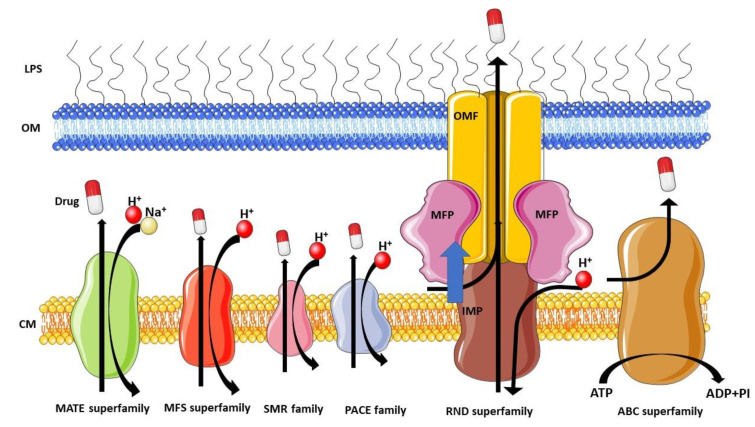
Schematic overview of the major families of MDR efflux pumps in *P. aeruginosa*. This figure was partly generated using Servier Medical Art, provided by Servier, licensed under a Creative Commons Attribution 3.0 unported license.

**Figure 2 antibiotics-12-01304-f002:**
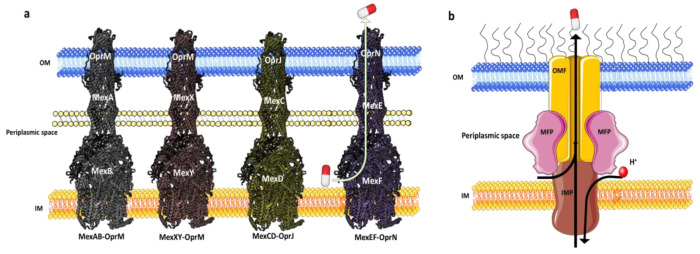
Multidrug efflux pumps in *P. aeruginosa*. (**a**) Schematic structures of the four major efflux pumps implicated in antibiotic resistance in *P. aeruginosa*, presenting the resistance-nodulation-cell division transporters (MexB, MexY, MexD, and MexF) on IM; the periplasmic membrane fusion proteins (MexA, MexX, MexC, and MexE) on the periplasm; and the channel-forming OM factors (OprM, OprJ, and OprN) on the OM. Protein descriptions are based on a protein databank (PDB) MexAB-OprM structure; PDB id; 6TA6 [137]. (**b**) Schematic overview of RND efflux pumps in *P. aeruginosa*. This figure was created using PDB data and was partly generated using Servier Medical Art, provided by Servier, licensed under a Creative Commons Attribution 3.0 unported license.

**Figure 4 antibiotics-12-01304-f004:**
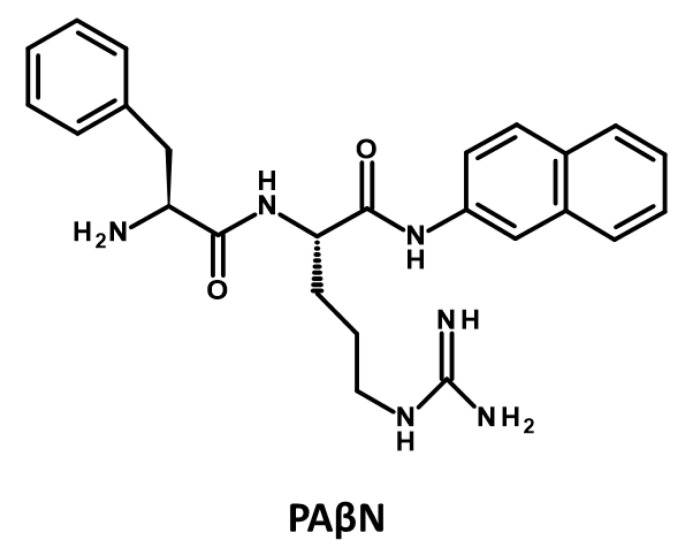
Structure of PAβN. This figure was prepared with BIOVIA Draw 2022.

**Figure 5 antibiotics-12-01304-f005:**
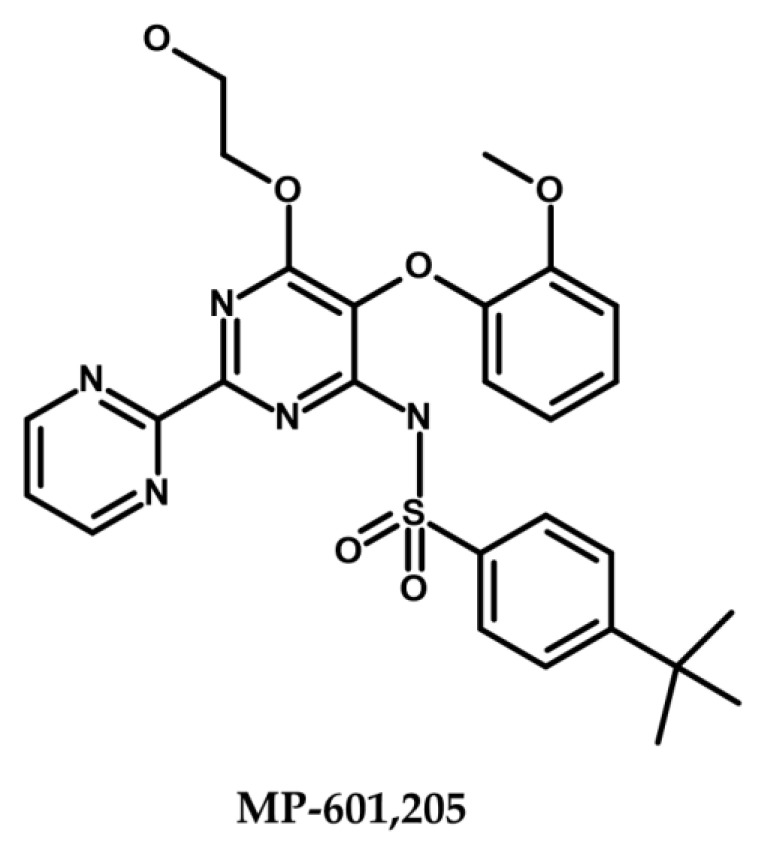
Structure of MP-601,205. This figure was prepared with BIOVIA Draw 2022.

**Figure 6 antibiotics-12-01304-f006:**
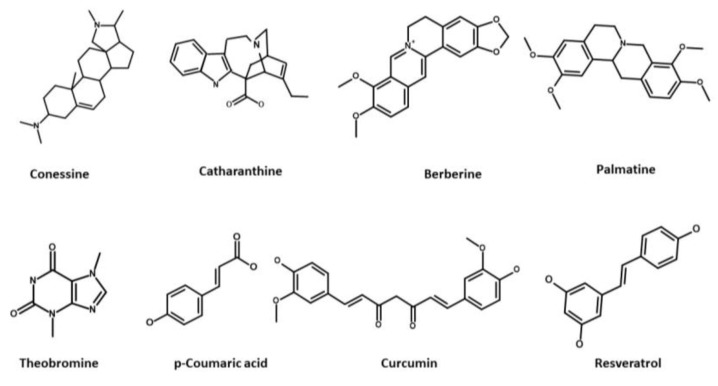
Chemical structures of potent plant-derived EPIs targeting Mex efflux pumps in *P. aeruginosa*. This figure was prepared with BIOVIA Draw 2022.

**Figure 7 antibiotics-12-01304-f007:**
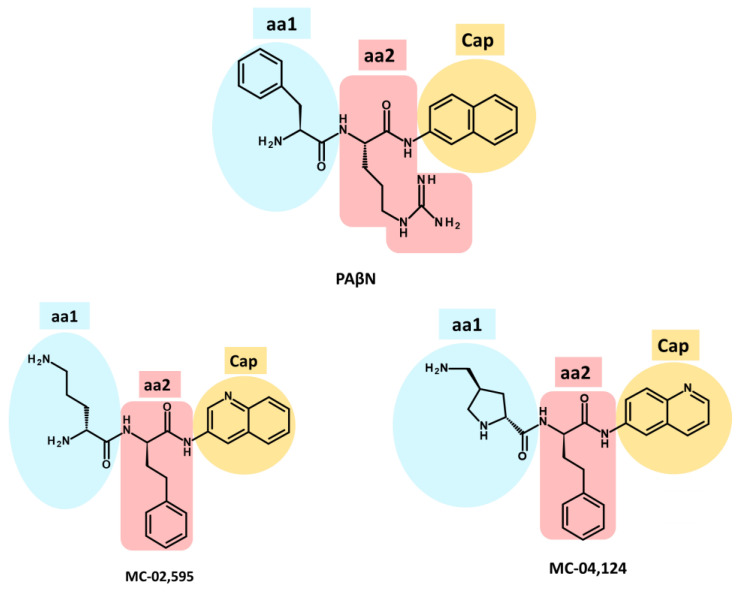
Structure of PAβN and its two major derivatives. aa: amino acid. This figure was prepared with BIOVIA Draw 2022. Adapted with permission from [136], 2023, Compagne et al. Licensed under Creative Commons Attribution 4.0 International License (CC-BY 4.0).

**Figure 8 antibiotics-12-01304-f008:**
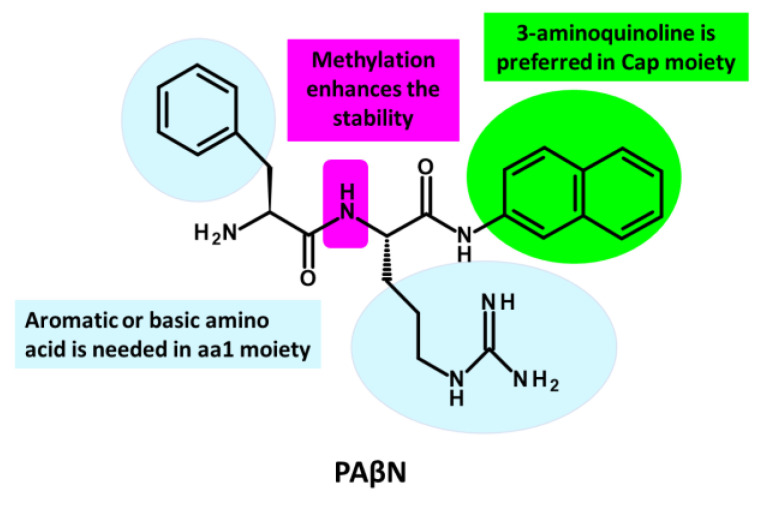
PAβN structure and SAR overview for enhancing its EPI activity. This figure was prepared with BIOVIA Draw 2022. Adapted with permission from [136], 2023, Compagne et al. Licensed under Creative Commons Attribution 4.0 International License (CC-BY 4.0).

**Figure 9 antibiotics-12-01304-f009:**
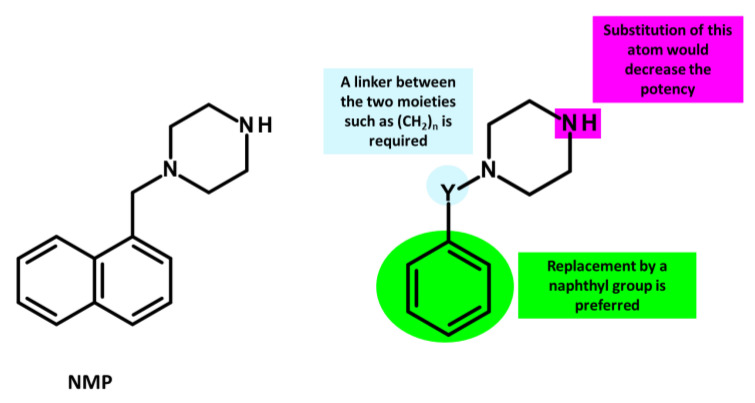
NMP structure and SAR overview for enhancing its EPI activity. This figure was prepared with BIOVIA Draw. Adapted with permission from [136], 2023, Compagne et al. Licensed under Creative Commons Attribution 4.0 International License (CC-BY 4.0).

**Figure 10 antibiotics-12-01304-f010:**
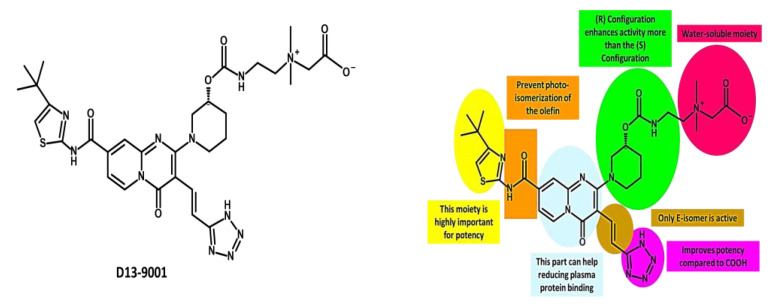
D13-9001 structure and SAR overview for enhancing its EPI activity. This figure was prepared with BIOVIA Draw. Adapted with permission from [136], 2023, Compagne et al. Licensed under Creative Commons Attribution 4.0 International License (CC-BY 4.0).

**Figure 11 antibiotics-12-01304-f011:**
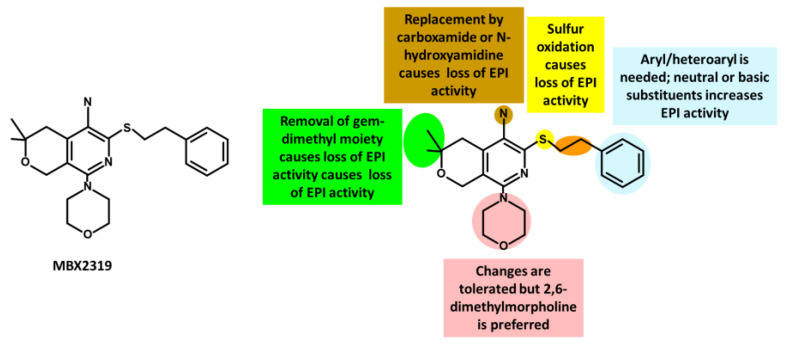
MBX2319 structure and SAR overview for enhancing its EPI activity. This figure was prepared with BIOVIA Draw. Adapted with permission from [136], 2023, Compagne et al. Licensed under Creative Commons Attribution 4.0 International License (CC-BY 4.0).

**Figure 12 antibiotics-12-01304-f012:**
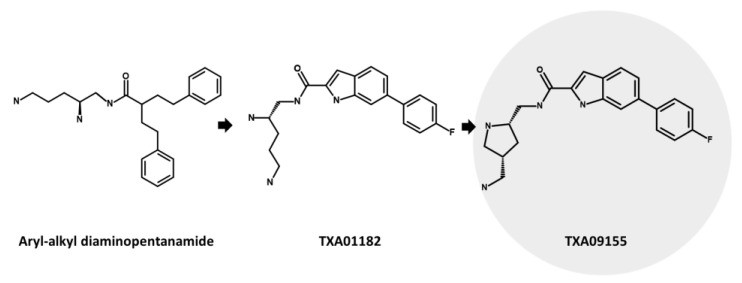
Structure of TXA09155, TXA01182, and aryl-alkyl diaminopentanamide analogue. This figure was prepared with BIOVIA Draw. The arrows in the figure show the lead optimization processes, highlighting the modifications made to the chemical structure to improve its drug-like properties, leading to the introduction of TXA09155 Adapted with permission from [136], 2023, Compagne et al. Licensed under Creative Commons Attribution 4.0 International License (CC-BY 4.0).

**Figure 13 antibiotics-12-01304-f013:**
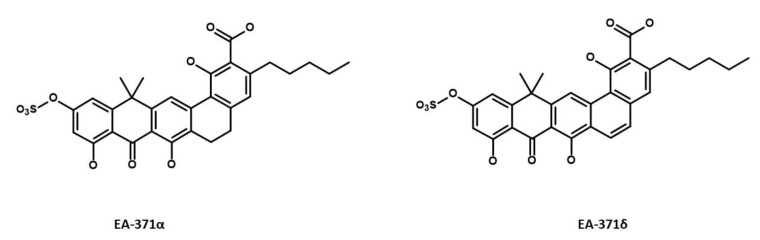
The EPIs EA-371α and EA-371δ, derived from a novel *Streptomyces* strain closely related to *Streptomyces vellosus*, appear to be potent EPIs against *P. aeruginosa*. This figure was prepared with BIOVIA Draw.

**Figure 14 antibiotics-12-01304-f014:**
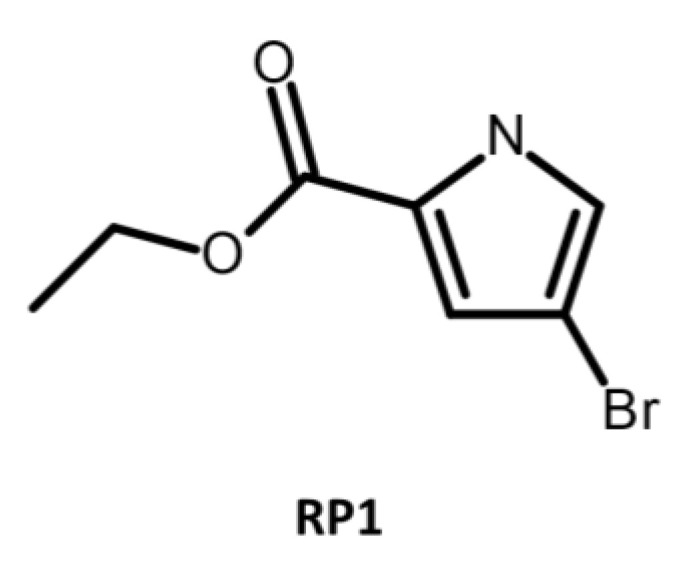
Structure of RP1, derived from the soil bacterium *Streptomyces* Sp., functions as an EPI targeting *P. aeruginosa*. This figure was prepared with BIOVIA Draw.

**Figure 15 antibiotics-12-01304-f015:**
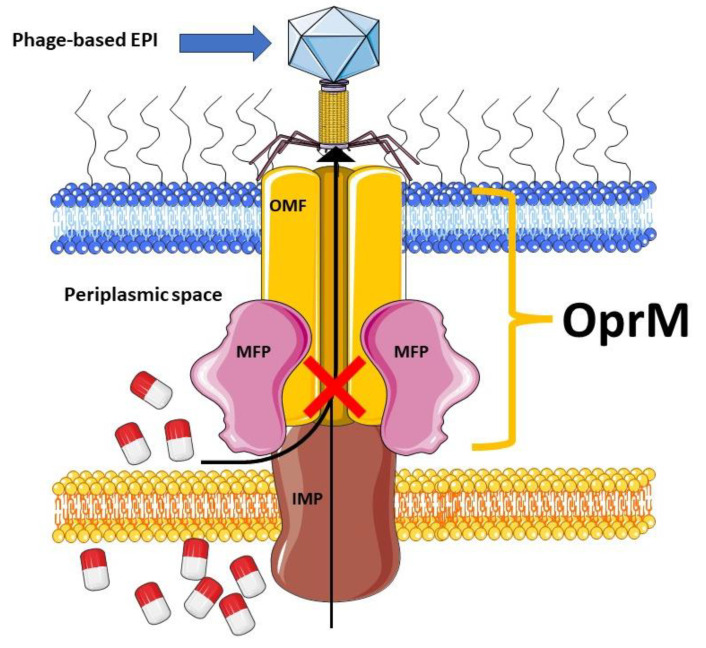
Schematic representation of the phage-based EPI approach using the outer efflux protein (OprM) as a receptor to block efflux. This leads to the accumulation of antibiotics in *P. aeruginosa*, offering a promising strategy to enhance antibiotic efficacy. This figure was partly generated using Servier Medical Art, provided by Servier, licensed under a Creative Commons Attribution 3.0 unported license.

**Figure 16 antibiotics-12-01304-f016:**
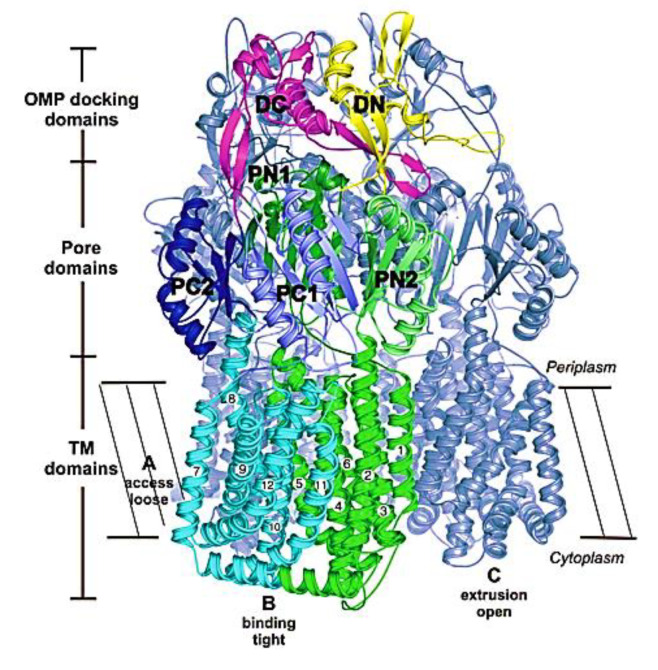
MexB’s structure and architecture are represented by a ribbon diagram of the trimer, showcasing its arrangement in the membrane plane. It highlights the three major domains and provides an approximate location of the IM. A monomer is color-coded to delineate its subdomains, with cyan representing one subdomain and navy blue and green indicating the others. The subdomains are labeled, and the α-helices are numbered from 1 to 12. Reprinted with permission from ref. [307], 2009, Sennhauser et al. This article is licensed under the Creative Commons Attribution license (CC-BY).

**Figure 17 antibiotics-12-01304-f017:**
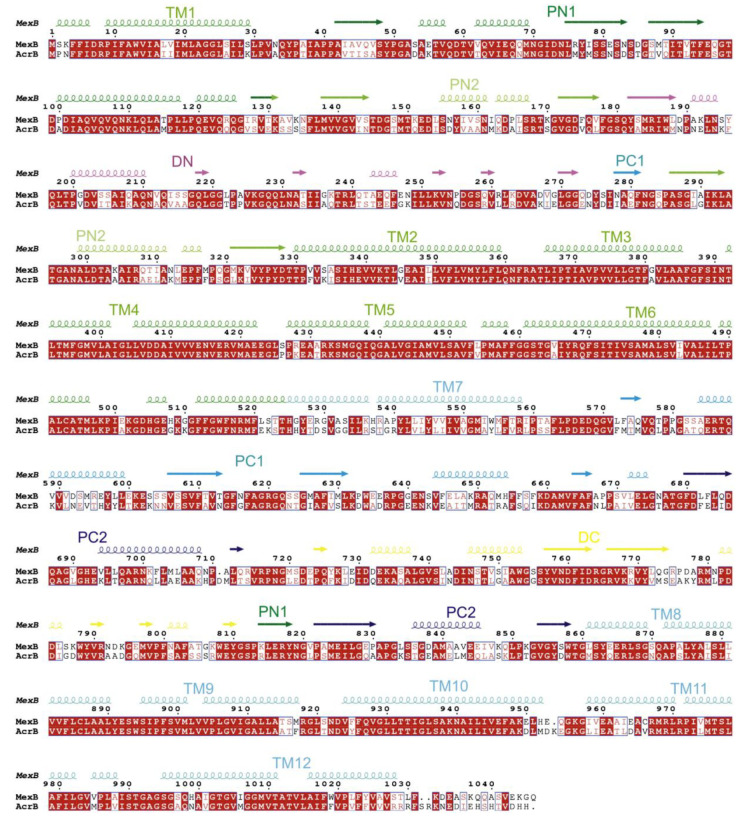
Alignment of MexB and AcrB sequences. Secondary structure elements are shown above the MexB sequence based on subunit C of the MexB crystal structure. Different subdomain colors align with those presented in Figure 16. Reprinted with permission from ref. [307], 2009, Sennhauser et al. This article is licensed under the Creative Commons Attribution license (CC-BY).

**Figure 18 antibiotics-12-01304-f018:**
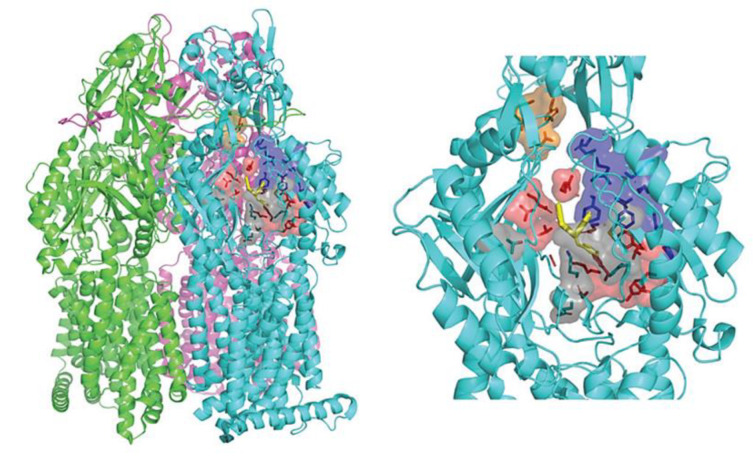
MexB structure: the left panel highlights L, T, and O monomers of MexB colored in green, cyan, and magenta, respectively (PDB Id: 3W9J). The detailed visualization on the right shows the different DP_T_ sub-regions of MexB: the interface is colored in gray (residues S79, T91, K134, F573, F617, M662, and E673), the cave is colored in red (residues Q46, T89, T130, N135, F136, V139, Q176, K292, Y327, V571, R620, and F628), and the groove is colored in blue (residues K151, F178, G179, R180, D274, S276, I277, A279, S287, P326, F610, V612, F615, and V47; S48, Q125, G126, R128, Q163, D174, F175, and Q273 that are located near the exit gate, colored in orange). The switch loop is represented as a yellow cartoon. Reprinted with permission from ref. [315], 2022, Gervasoni et al. This article is licensed under a Creative Commons Attribution-Noncommercial 3.0 unported license (CC BY-NC 3.0).

**Figure 19 antibiotics-12-01304-f019:**
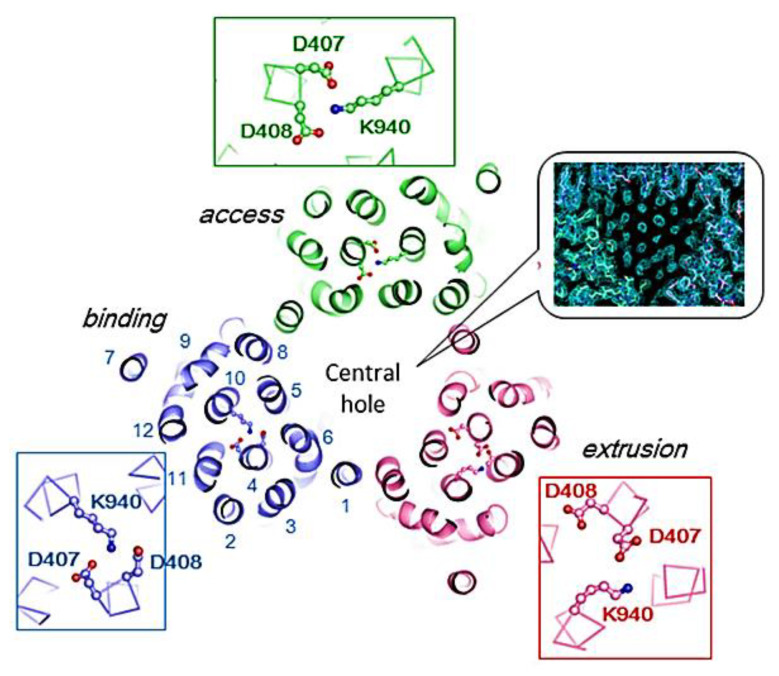
Horizontal cut view of the transmembrane region of the asymmetric AcrB trimer. Asp407, 408, and Lys940, which form ion pairs in the transmembrane core region, are depicted as ball-and-stick models. The inset shows the electron density observed in the transmembrane hole at the center of the MexB transmembrane trimers, indicating that the hole is filled with a phospholipid bilayer. The protein structures are depicted using ribbon models, with green, blue, and pink representing the access, binding, and extrusion monomers, respectively. Bound drugs are illustrated as stick models. Reprinted with permission from [61], 2015, Yamaguchi et al. This article is licensed under the Creative Commons Attribution license (CC-BY).

**Figure 20 antibiotics-12-01304-f020:**
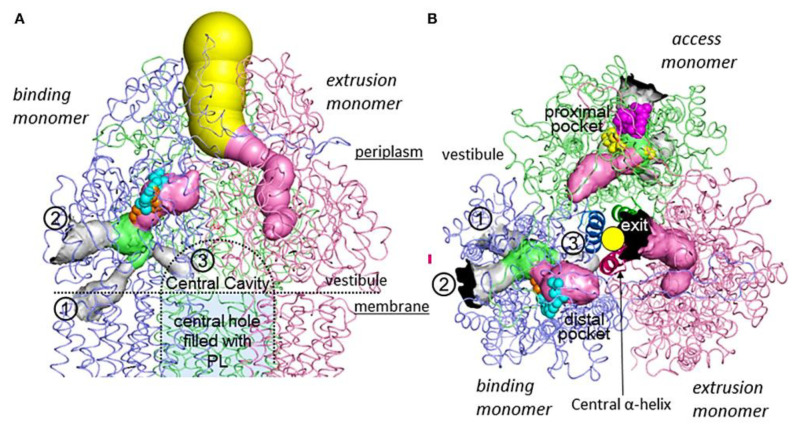
Intramolecular water-accessible channels in the AcrB trimer. The proximal pocket, distal pocket, entrances, and funnel-like exit are depicted in green, pink, gray, and yellow, respectively. The channel apertures at the entrance and exit are depicted in black. (1) IM entrance, (2) periplasmic entrance, (3) central cavity entrance. (**A**) Side view. The central cavity and central hole are depicted as dotted lines. (**B**) Horizontal cut view of the porter domain. The yellow circle indicates the closed pore-like structure comprising three central α-helices (depicted as a ribbon model with dense color), which was postulated to be a part of the putative substrate translocation channel during the early stages. Bound minocycline (cyan), doxorubicin (orange), rifampicin (magenta), and erythromycin (yellow) overlap in the space-filling model.

**Figure 21 antibiotics-12-01304-f021:**
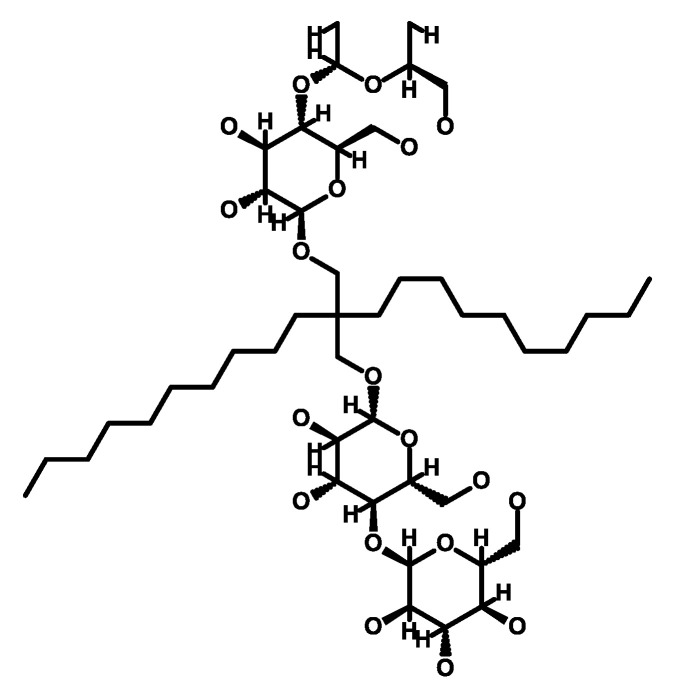
Chemical structure of lauryl maltose neopentyl glycol (LMNG). This figure was prepared with BIOVIA Draw.

**Figure 22 antibiotics-12-01304-f022:**
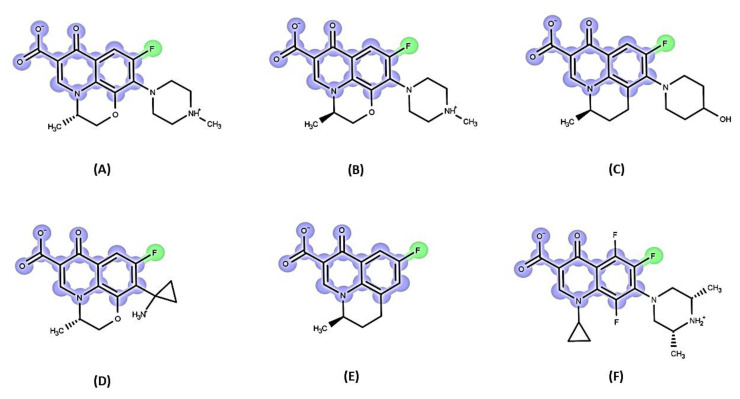
Examples of fluoroquinolone (FQ) structures: (**A**) levofloxacin, (**B**) ofloxacin, (**C**) nadifloxacin, (**D**) pazufloxacin, (**E**) flumequine, and (**F**) orbifloxacin. The molecular scaffold shared by all quinolones is highlighted in blue, while the presence of fluorine atoms, characteristic of FQs, is indicated in green. Adapted with permission from ref. [315], 2022, Gervasoni et al. This article is licensed under a Creative Commons Attribution-Noncommercial 3.0 unported license (CC BY-NC 3.0).

**Figure 23 antibiotics-12-01304-f023:**
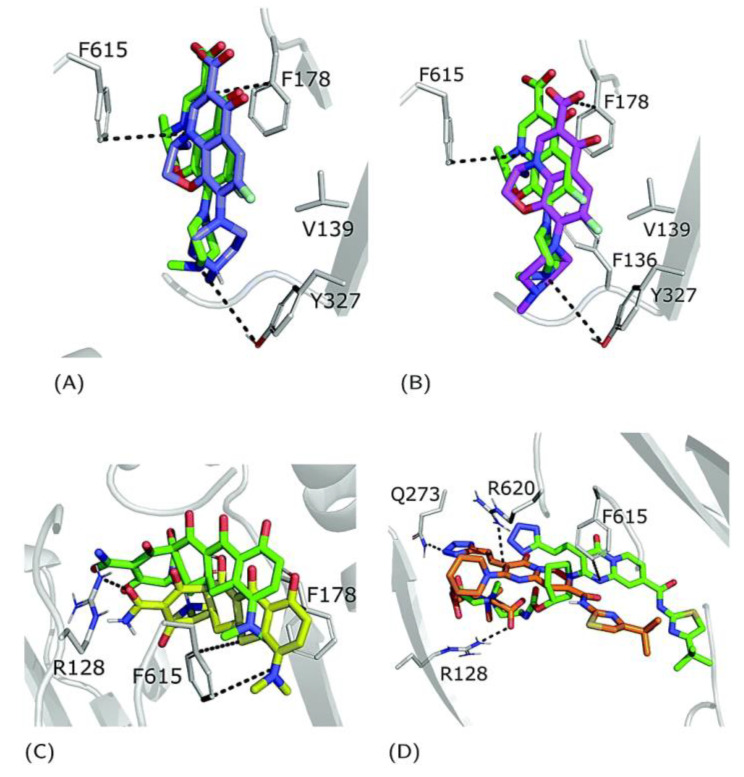
Comparison between the crystal structure (colored in green) and the docking poses of the substrate localizations trapped inside MexB: (**A**) zwitterionic levofloxacin in violet, (**B**) non-zwitterionic levofloxacin (net charge-1) in magenta, (**C**) minocycline in yellow, and (**D**) D13-9001 in orange. Levofloxacin and minocycline are in a complex with AcrB (PDB Ids: 7B8T32 and 4DX5,45, respectively), while D13-9001 is in a complex with MexB (PDB Id 3W9J29). Protein–ligand interactions are represented as dotted lines. Reprinted with permission from ref. [315], 2022, Gervasoni et al. This article is licensed under a Creative Commons Attribution-Noncommercial 3.0 unported license (CC BY-NC 3.0).

**Figure 24 antibiotics-12-01304-f024:**
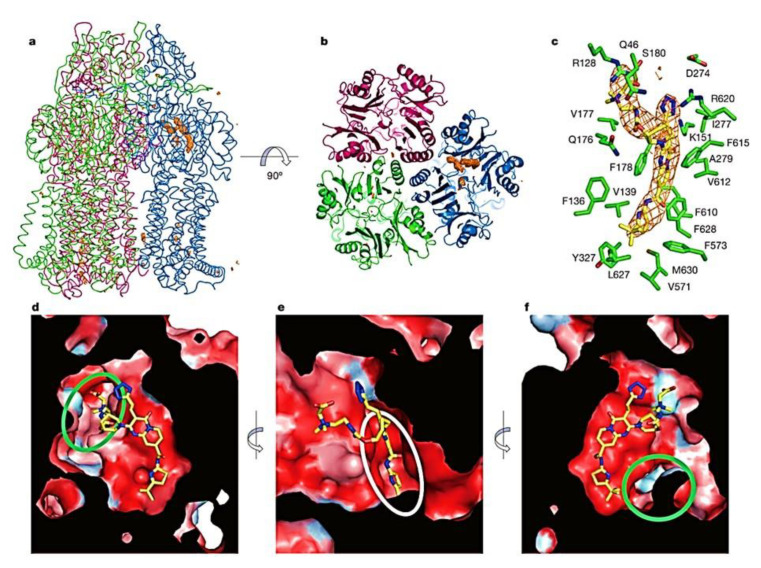
Crystal structure of the MexB trimer complexed with the EPI D13-9001. This figure provides several key views, including the overall structure of the D13-9001-bound MexB trimer (**a**), a top-down cutaway view of the head-piece region in the D13-9001-bound MexB structure (**b**), a close-up view highlights the precise binding site of D13-9001 within MexB (**c**), and cutaway views of the distal drug-binding pocket from different angles: towards the exit ((**d**); green oval), looking down the hydrophobic trap from the substrate-translocation channel ((**e**); white oval), and towards the entrance ((**f**); green oval). Reprinted with permission from ref. [318], 2013, Nakashima et al. This article is licensed under the Creative Commons Attribution license (CC-BY).

**Figure 25 antibiotics-12-01304-f025:**
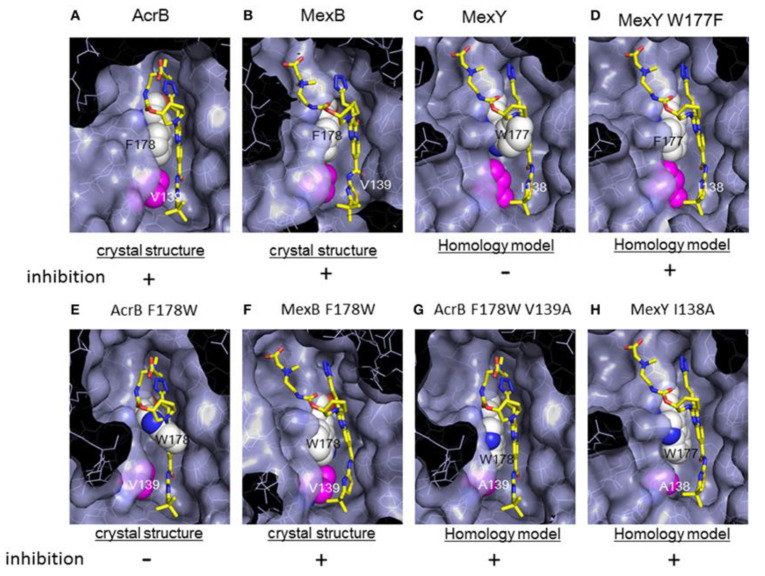
A magnified view of the D13-9001 binding site is depicted as a surface model. D13-9001 is shown in stick form, F178 and W177 in white space-filling models, and V139, I138, and mutated Ala in magenta space-filling models. The symbols + and - indicate inhibition or the lack of inhibition by D13-9001. Crystal structures (**A**,**B**,**E**,**F**) and homology models (**C**,**D**,**G**,**H**) illustrate the binding of D13-9001 to AcrB, MexB, MexY, MexY W177F, AcrB F178W, ABI-PP-binding MexAB F178W, AcrB F178W V139A, and MexY I138A. Reprinted with permission from [61], 2015, Yamaguchi et al. This article is licensed under the Creative Commons Attribution license (CC-BY).

**Table 1 antibiotics-12-01304-t001:** Major families of bacterial efflux pumps in *P. aeruginosa* and their substrate specificities.

Family ^1^	Efflux Pump	Gene Ids ^2^	Substrates ^3^	References
**ABC**	Ttg2 (Mla)	PA4456-PA4455-PA4454-PA4453-PA4452	CHL, CIP, COL, DMF, DOX, LVX, MIN, OFX, TET, TGC, TOB, TMP	[80,81,82]
	PA1874-77	PA1874-PA1875-PA1876-PA1877	CIP, GEN, NOR, TOB	[83,84]
	PA3228	PA3228	CAR, LVX, NOR	[85]
**MFS**	Mfs1	PA1262	PQT	[86]
	Mfs2/SmvA	PA1282	OCT, PQT	[86,87]
	CmlA1	GNT62_RS22140	CHL	[88,89]
**SMR**	PASmr/EmrEPae	PA4990	ACR, EtBr, GEN, KAN, NEO	[90,91]
	SugE subfamily SMR	PA1882	Further research is needed	[92]
**MATE**	PmpM	PA1361	ACR, BZK, CIP, EtBr, NOR, OFX, TPPCL	[93]
**PACE**	PA2880	PA2880	CHX	[94]
**RND**	MexAB-OprM	PA0425-PA0426-PA0427	AMI, AMX, ATM, CAR, CR, MA, FEP, CFP, CFSL, CTX, FOX, CZOP, CPO, CES, CAZ, CZX, CRO, CXM, CHL, CTET, CIN, CIP, CLX, DP, DOR, ENX, ERY, FMOX, GEN, IPM, LVX, CLM, MEM, MOX, NAF, NAL, NOR, NOV, OFX, OMC, OTC, PG, PPA, PIP, PTZ, PMA, SPX, SPI, STN, SUL, TZB, TET, TIC, TOS	[95,96,97,98]
	MexCD-OprJ	PA4599-PA4598-PA4597	AMX, MA, FEP, CFP, CFSL, CTX, FOX, CZOP, CPO, CES, CZX, CRO, CXM, CHL, CHX, CTET, CIN, CIP, CLX, DOR, ENX, ERY, FMOX, LVX, CLM, MEM, NAF, NAL, NOR, NOV, OFX, OMC, OTC, PG, PPA, PIP, PMA, SPX, SPI, TET, TOS	[96,97,99,100]
	MexCD-TOprJ	LSG45_RS29735-LSG45_RS29740-LSG45_RS29745	FEP, CEQ, CAZ, CTET, CIP, DOX, ERV, FLO, GEN, MIN, NAL, OTC, STR, TET, TGC	[101,102]
	MexEF-OprN	PA2493-PA2494-PA2495	CHL, QN, TET, TMP	[103]
	MexGHI-OpmD	PA4205-PA4206-PA4207-PA4208	5-Me-PCA, ACR, EtBr, NOR, R6G, TET, V	[104,105,106]
	MexJK-OprM	PA3677-PA3676-PA0427	ERY, TET, TCS	[107]
	MexMN-OprM	PA1435-PA1436-PA0427	BAL30072, ATM, BIPM, CAR, CMN, CAZ, CFT, CHL, MEM, MET, MOX, NOV, PIP, SUL, TMC, TP, TIC	[108,109]
	MexPQ-OpmE	PA3523-PA3522-PA3521	Hoechst 33342, CHL, CIP, ERY, KIT, NOR, RKM, TET, TPPCL	[108]
	MexVW-OprM	PA4374-PA4375-PA0427	ACR, CPO, CHL, ERY, EtBr, NOR, OFX, TET	[110]
	MexXY-OprM (-OprA)	PA2019-PA2018-PA0427 (-PSPA7_3271)	ACR, AMI, AMX, MA, FEP, CFP, CFSL, CTX, FOX, CZOP, CPO, CAZ, CZX, CRO, CXM, CHL, CTET, CIN, CIP, CLX, DOR, ENX, ERY, EtBr, FMOX, GEN, IPM, KAN, LVX, CLM, MEM, NAF, NAL, NEO, NOR, OFX, OMC, OTC, PG, PPA, PIP, PTZ, PMA, SPX, SPI, STR, TZB, TET, TIC, TOB, TOS, CAR (only with OprA), SUL (only with OprA)	[96,97,111,112]
	MuxABC-OpmB	PA2528-PA2527-PA2526-PA2525	ATM, ERY, KIT, NOV, RKM, TET	[113]
	TriABC-OpmH	PA0156-PA0157-PA0158-PA4974	TCS	[114]

^1^: ABC: ATP-binding cassette transporters; MFS: major facilitator superfamily; SMR: small multidrug resistance; MATE: multidrug and toxic-compound extrusion; PACE: proteobacterial antimicrobial compound efflux; RND: resistance-nodulation-cell division. ^2^: IDs obtained from the *Pseudomonas* Genome Database [115]. ^3^: ACR: acriflavine; AMI: amikacin; AMX: amoxicillin; ATM: aztreonam; BIPM: biapenem; BZK: benzalkonium chloride; CAR: carbenicillin; CAZ: ceftazidime; CEQ: cefquinome; CES: cefsulodin; CFP: cefoperazone; CFSL: cefoselis; CFT: ceftolozane; CHL: chloramphenicol; CHX: chlorhexidine; CIN: cinoxacin; CIP: ciprofloxacin; CLX: cloxacillin; CMN: carumonam; COL: colistin; CPO: cefpirome; CR: carvacrol; CRO: ceftriaxone; CTET: chlortetracycline; CTX: cefotaxime; CXM: cefuroxime; CZOP: cefozopran; CZX: ceftizoxime; DMF: dimethylformamide; DOR: doripenem; DOX: doxycycline; DP: dipyridyl; ENX: enoxacin; ERV: eravacycline; ERY: erythromycin; EtBr: ethidium bromide; FEP: cefepime; FLO: florfenicol; FMOX: flomoxef; FOX: cefoxitin; GEN: gentamicin; IPM: imipenem; KAN: kanamycin; KIT: kitasamycin; LCM: lincomycin; LVX: levofloxacin; MA: cefamandole; MEM: meropenem; MET: methicillin; MIN: minocycline; MOX: moxalactam; NAF: nafcillin; NAL: nalidixic acid; NEO: neomycin; NOR: norfloxacin; NOV: novobiocin; OCT: octenidine; OFX: ofloxacin; OMC: oleandomycin; OTC: oxytetracycline; PG: penicillin G; PIP: piperacillin; PMA: piromidic acid; PPA: pipemidic acid; PQT: paraquat; PTZ: piperacillin-tazobactam; QN: quinolones; R6G: rhodamine 6G; RKM: rokitamycin; SPI: spiramycin; SPX: sparfloxacin; STN: streptonigrin; STR: streptomycin; SUL: sulbenicillin; TCS: triclosan; TET: tetracycline; TGC: tigecycline; TIC: ticarcillin; TMC: temocillin; TMP: trimethoprim; TOB: tobramycin; TOS: tosufloxacin; TP: thiamphenicol; TPPCL: tetraphenylphosphonium chloride; TZB: tazobactam; V: vanadium. Adapted with permission from ref. [116], 2022, Lorusso et al. Licensed under Creative Commons Attribution 4.0 International License (CC-BY 4.0).

## Data Availability

Not applicable.

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
