# Peer review of "The Art of War with Pseudomonas aeruginosa: Targeting Mex Efflux Pumps Directly to Strategically Enhance Antipseudomonal Drug Efficacy"

_antibiotics, 2023, doi:10.3390/antibiotics12081304_

Round 1
Reviewer 1 Report
Pseudomonas aeruginosa is a highly drug-resistant bacterium that poses a significant clinical challenge, leading to severe infections. This publication emphasizes the importance of addressing multidrug resistance (MDR) in P. aeruginosa, specifically focusing on the Mex efflux pumps and the potential of efflux pump inhibitors (EPIs) to restore antibacterial activity by countering or bypassing efflux activities. While minor revisions and clarifications may be necessary to improve the clarity and flow of the text.
Lane 42: Please provide the references.
Lane 62-63: It would be helpful to provide more detailed explanations and examples, particularly regarding the genes and enzymes involved in order to enhance understanding.
Lane 93-98: Please provide the references.
Lane 161: "Inner manbrane" should be corrected to "inner membrane."
Lane 158-159: Please provide references to support the statement.
Lane 199-200: Consider revising the sentence "Efflux pumps served as a vital tool for bacteria to interact with the environment, playing a crucial role in their evolutionary journey" to provide more clarity on how efflux pumps contribute to bacterial evolution and interaction with the environment. Elaborate on their specific mechanisms and impacts.
Lane 250-251: Please provide references.
Lane 256, 289: The numbering should be corrected to 2.1 instead of 3.1, and so on for the rest of the numbering.
Lane 268: Two commas after citation 137.
Lane 313-316: In the sentence "Notably, specific amino acid substitutions in MexY have been associated with an enhanced resistance profile against aminoglycosides, cefepime, and FQs," consider specifying the nature of the amino acid substitutions and their impact on resistance in order to provide more clarity.
Lane 382-385: Please provide references.
Lane 482: In section 4, correct the numbering of the subtitles.
Lane 498-499: Please provide references. Clarify the relationship between EPIs and biofilm formation. Explain how EPIs interfere with the efflux pumps involved in biofilm formation and how this disruption affects the attachment and growth of bacteria in biofilms. Elaborate on the specific processes that EPIs target and their impact on antibacterial tolerance within biofilms to strengthen the readers' understanding.
Lane 562-572: Expand on the various mechanisms by which EPIs can exert inhibitory effects on efflux pump activities. Provide more specific examples and references to support each mechanism, including competitive inhibition, non-competitive substrate action, hindering functional movement, and co-substrate action. This will enhance the understanding of EPIs' diverse mechanisms of action. L
ane 590-591: Please provide references.
Lane 600-602: Please provide references.
Lane 699: Please correct the reference.
Lane 788-792: This is a repeated paragraph from lane 793-799.
Lane 875-876: Please provide references.
Lane 881-887: Please provide references.
Lane 903-904: Please provide references.
Lane 1227-1241: Please provide more specific references.
Lane 1258: There is a lack of a period.
Section 6.3: Provide specific references to support the statements and claims made throughout the section.
Discussion:
The discussion comprehensive, informative, and appropriately address the relevant aspects of efflux pumps and EPIs in P. aeruginosa strains. Consider the following minor points for further improvement:
- Expand on the challenges posed by membrane porins and their selectivity for smaller hydrophilic molecules. Discuss the rationale behind incorporating charged groups into the structure of EPIs to facilitate interaction with the lipid bilayer and overcome the limitations of porins. Highlight the need for innovative strategies to bridge the molecular dichotomy between membrane porins and efflux pumps.
- Emphasize the importance of computational studies, in silico modeling, and biophysical techniques in understanding the structure, dynamics, and molecular interactions of EPIs with efflux pumps. Discuss the role of crystal structures, homology models, native mass spectrometry, and NMR in providing insights into the binding sites and conformational changes involved in efflux pump inhibition. Highlight the potential of structure-based approaches in the discovery and characterization of novel EPIs.
- Provide more information on the specific challenges faced in developing EPIs that possess both hydrophilic and hydrophobic characteristics to strike a balance between penetrating the outer membrane and interacting with efflux pumps.
- Consider expanding on the potential strategies for bypassing or circumventing the outer membrane barrier while effectively inhibiting efflux pumps. Provide more insights or examples of these innovative approaches.
Reviewer 2 Report
I think this work could be improve if the author add some points as below:
1. Prioritizing the assessment of the ADMET (absorption, distribution, metabolism, excretion, and toxicity) profile of the EPI (efflux pump inhibitor) before proceeding to clinical trials. This would help ensure the safety and efficacy of the EPI in a clinical setting.
2. Conducting structure-activity relationship (SAR) studies to identify and address challenges related to permeability and solubility. This would help optimize the properties of the EPI and improve its effectiveness.
3. Exploring alternative approaches to traditional in vitro and in vivo assays for studying EPIs and their interactions with efflux pumps. Computational studies and complementary biophysical techniques, such as in silico modeling and crystal structures, can provide valuable insights into the design and mechanisms of action of EPIs.
4. Considering specific characteristics for successful EPIs, such as a molecular weight of 300 or less, a lipophilicity value (clogP) of 3 or lower, a polar surface area (PSA) of 3 or less, a maximum of three rotatable bonds, and no more than three hydrogen bond donors and acceptors. Focusing on these criteria during the screening process can help identify potential EPIs with desired properties for efficient inhibition of efflux pumps.
